# Plexins promote Hedgehog signaling through their cytoplasmic GAP activity

Justine M Pinskey[1][†], Tyler M Hoard[1][†], Xiao-Feng Zhao[1], Nicole E Franks[1], Zoë C Frank[1], Alexandra N McMellen[1], Roman J Giger[1,2], Benjamin L Allen[1]*

[1]Department of Cell and Developmental Biology, University of Michigan, Ann Arbor, United States; [2]Department of Neurology, University of Michigan, Ann Arbor, United States

**Abstract** Hedgehog signaling controls tissue patterning during embryonic and postnatal development and continues to play important roles throughout life. Characterizing the full complement of Hedgehog pathway components is essential to understanding its wide-ranging functions. Previous work has identified neuropilins, established semaphorin receptors, as positive regulators of Hedgehog signaling. Neuropilins require plexin co-receptors to mediate semaphorin signaling, but the role of plexins in Hedgehog signaling has not yet been explored. Here, we provide evidence that multiple plexins promote Hedgehog signaling in NIH/3T3 mouse fibroblasts and that plexin loss of function in these cells results in significantly reduced Hedgehog pathway activity. Catalytic activity of the plexin GTPase-activating protein (GAP) domain is required for Hedgehog signal promotion, and constitutive activation of the GAP domain further amplifies Hedgehog signaling. Additionally, we demonstrate that plexins promote Hedgehog signaling at the level of GLI transcription factors and that this promotion requires intact primary cilia. Finally, we find that plexin loss of function significantly reduces the response to Hedgehog pathway activation in the mouse dentate gyrus. Together, these data identify plexins as novel components of the Hedgehog pathway and provide insight into their mechanism of action.

*For correspondence:
benallen@umich.edu

[†]These authors contributed equally to this work

Competing interest: The authors declare that no competing interests exist.

## Editor's evaluation

This work demonstrates that Plexins, like their neuropilin-binding partners, promote HH signaling. The authors use both in vitro signaling assays, knockdown in chick neural tube patterning assays and PlexinA1 and A2 mutant mice to demonstrate that several Plexins enhance HH signaling in a way that depends on the Plexin GAP domain.

## Introduction

The Hedgehog (HH) signaling pathway utilizes a core set of components to coordinate diverse cellular processes. In the absence of HH ligand, the 12-pass transmembrane (TM) protein Patched 1 (PTCH1) inhibits pathway activity by repressing a second cell-surface protein Smoothened (SMO), a 7-pass TM protein with GPCR-like activity (*Alcedo et al., 1996*; *Marigo and Tabin, 1996*; *Stone et al., 1996*; *van den Heuvel and Ingham, 1996*). HH ligand binding to PTCH1 leads to de-repression of SMO, which shifts the processing of GLI transcription factors from repressor to activator forms, thus altering the balance of HH target gene expression (*Hui and Angers, 2011*). By balancing the activity of these key molecules, HH signaling directs embryonic and postnatal development as well as adult tissue homeostasis in a wide variety of cellular contexts. In contrast, HH pathway disruption can drive a number of diseases, including cancer (*Teglund and Toftgård, 2010*; *Briscoe and Thérond, 2013*; *Petrova and Joyner, 2014*).

Beyond these core pathway components, a growing number of additional proteins regulate HH signaling at the cell surface in a tissue- and stage-specific manner (*Beachy et al., 2010*). Notably, many of these cell surface regulators have partially redundant functions (*Zhang et al., 2001*; *Jeong and McMahon, 2005*; *Allen et al., 2007*; *Allen et al., 2011*; *Izzi et al., 2011*; *Holtz et al., 2013*). Further, increased complexity within vertebrate HH signaling, including a requirement for the primary cilium, has made it difficult to study HH regulators that lack invertebrate counterparts, such as Scube2 and GAS1 (*Ingham et al., 2011*; *Wierbowski et al., 2020*). Therefore, our overall understanding of vertebrate HH signaling remains incomplete.

The semaphorins (SEMA) are a large family of membrane-bound and secreted proteins that regulate cell migration, axon guidance, synapse assembly, angiogenesis, immune function, and cell death (*Yazdani and Terman, 2006*; *Jongbloets and Pasterkamp, 2014*; *Koropouli and Kolodkin, 2014*; *Fard and Tamagnone, 2021*). Neuropilins (NRPs) directly interact with class 3 SEMA ligands and require plexin (PLXN) co-receptors to transduce SEMA signals intracellularly (*Chen et al., 1997*; *He and Tessier-Lavigne, 1997*; *Kolodkin et al., 1997*; *Takahashi et al., 1999*; *Tamagnone et al., 1999*; *Gu et al., 2005*). Membrane-bound SEMA and Sema3E interact directly with PLXN extracellular domains (ECDs) to activate downstream signaling events, which lead to remodeling and disassembly of the cytoskeleton (*Barberis et al., 2004*; *Neufeld and Kessler, 2008*; *Jongbloets and Pasterkamp, 2014*; *Rich et al., 2021*). PLXNs are a family of conserved, single-pass TM proteins containing nine different receptor types, which fall into four subfamilies based on homology (A, B, C, and D) (*Tamagnone et al., 1999*). The cytoplasmic domain (CD) of all PLXN family members harbors a GTPase-activating protein (GAP) domain (*Rohm et al., 2000b*; *Wang et al., 2012*). Catalytic activity of the PLXN GAP domain is necessary for SEMA-mediated cytoskeletal remodeling and cell migration (*Hota and Buck, 2012*; *Wang et al., 2013*; *Zhao et al., 2018*). Importantly, there is a mechanistic link between HH and NRPs. Multiple lines of evidence show that NRPs positively regulate HH signaling through their CDs (*Ge et al., 2015*; *Hillman et al., 2011*; *Pinskey et al., 2017*); however, the role of PLXNs in HH signaling remains unexplored.

Here, we investigated the role of PLXNs in HH pathway regulation. Our data suggest that multiple PLXNs, including members of the PLXN A and B subfamilies, positively regulate HH signaling. Similar to NRPs, we find that the PLXN CD is necessary for HH regulation. Interestingly, while the mechanism of NRP action in HH signaling may diverge from its mechanism in SEMA signaling (*Andreyeva et al., 2011*; *Ge et al., 2015*; *Pinskey et al., 2017*), we discover that PLXNs function similarly in SEMA and HH cascades. Mutating key residues within the cytoplasmic PLXN GAP domain prevents PLXN from promoting HH signaling. Further, deleting the PLXN ECD to create a constitutively active receptor augments HH promotion and alters HH-dependent tissue patterning and cell migration in the embryonic chicken neural tube, suggesting that PLXNs positively regulate HH signaling through GAP enzymatic activity. Additionally, we determine that PLXNs act at the level of the GLI transcription factors, and that PLXNs require intact primary cilia to promote HH pathway activity. In the developing mouse hippocampus, we observe PLXN-dependent regulation of HH target gene expression in the dentate gyrus, in vivo. Taken together, these data identify PLXNs as novel components of the HH pathway and contribute to our mechanistic understanding of HH regulation at the cell surface.

## Results

### Multiple Plxns promote HH signaling in NIH/3T3 fibroblasts

PLXNs consist of nine members that can be classified into four different subfamilies based on homology (PLXNA1-4, PLXNB1-3, PLXNC1, and PLXND1) (*Tamagnone et al., 1999*; *Neufeld and Kessler, 2008*). PLXNs from the A and D subfamilies interact with NRP co-receptors (*Takahashi et al., 1999*; *Neufeld and Kessler, 2008*), which have been previously identified as positive regulators of HH signaling (*Hillman et al., 2011*; *Ge et al., 2015*; *Pinskey et al., 2017*). We initially investigated whether *Plxna1* expression in HH-responsive NIH/3T3 fibroblasts would impact HH signaling using a luciferase reporter assay (*Nybakken et al., 2005*; *Figure 1A*). Strikingly, and similar to what we previously observed with *Nrp1* (*Pinskey et al., 2017*), *Plxna1* expression significantly increases HH pathway activation compared to a vector-transfected (pCIG) control (*Figure 1B*). Of note, PLXNA1 does not promote HH signaling in the absence of pathway activation with HH ligand (*Figure 1B*). To address whether HH promotion was specific to PLXNA1, we also examined PLXNA2, PLXNA3, and PLXNA4.

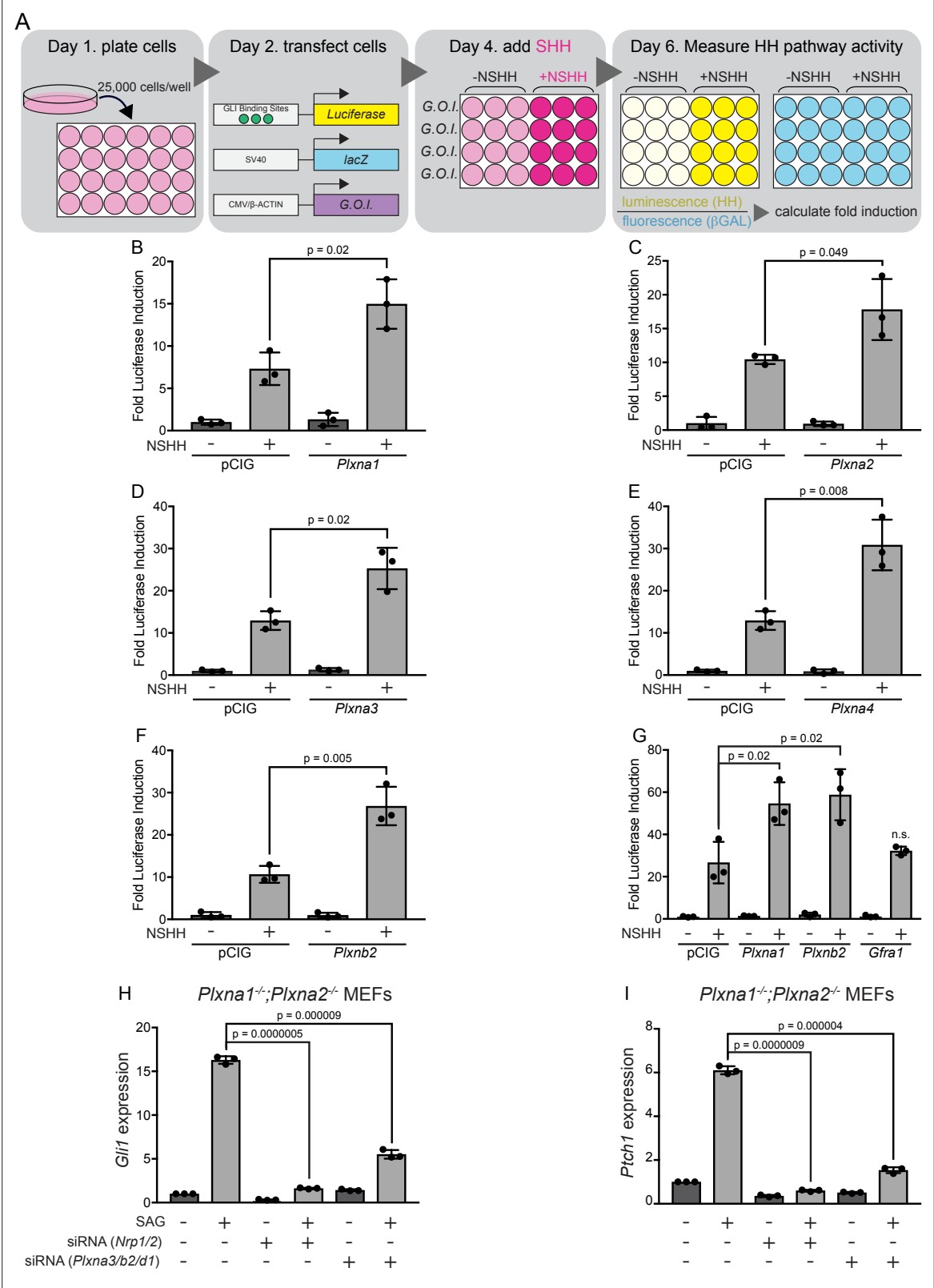

**Figure 1.** Multiple plexins (PLXNs) promote Hedgehog (HH) signaling. (**A**) Schematic of HH-responsive NIH/3T3 luciferase assays. G.O.I., gene of interest. (**B–F**) HH-dependent luciferase reporter activity was measured in NIH/3T3 cells transfected with the indicated constructs or empty vector control (pCIG) and stimulated with control (-NSHH) or NSHH-conditioned media (+NSHH). (**G**) Direct analysis of PLXNA1- and PLXNB2-mediated HH pathway promotion, compared with the unrelated cell surface protein GFRα1. (**H, I**) qRT-PCR analysis of *Gli1* and *Ptch1* in response to HH pathway

*Figure 1 continued on next page*

*Figure 1 continued*

activation via the Smoothened agonist, SAG. *Plxna1⁻/⁻;Plxna2⁻/⁻* mouse embryonic fibroblasts (MEFs) were treated with siRNA oligos for either *Nrp1* and *Nrp2* or *Plxna3, Plxnb2,* and *Plxnd1,* as indicated. Data points indicate technical replicates. Fold changes were determined using the ΔΔCT method. Data are reported as mean fold induction ± SD, with p-values calculated using two-tailed Student's *t*-tests. n.s., not significant.

The online version of this article includes the following source data and figure supplement(s) for figure 1:

**Source data 1.** Raw data for *Figure 1B–I*.

**Figure supplement 1.** *Plxn* expression in NIH/3T3 fibroblasts.

Our data suggest that all members of the PLXN A subfamily promote HH signaling following pathway activation with HH ligand (*Figure 1C–E*). We extended our analyses to include PLXNB2, which is not known to interact with NRPs (*Neufeld and Kessler, 2008*). Surprisingly, PLXNB2 also promotes HH signaling to a similar extent as PLXNs from the A subfamily, suggesting that PLXN-mediated HH promotion may be independent of NRP interaction (*Figure 1F and G*). Importantly, GFRα1, an unrelated cell-surface protein within the glial cell line-derived neurotrophic factor receptor (GFR) family, does not promote HH signaling (*Figure 1G*). Taken together, these data suggest that multiple PLXN family members promote HH signaling in NIH/3T3 cells.

## Plxn knockdown decreases HH-responsiveness in NIH/3T3 fibroblasts

According to RNA-sequencing data from the ENCODE project (*ENCODE Project Consortium, 2012*; *Davis et al., 2018*), NIH/3T3 fibroblasts express a subset of Plxns at varying levels, with *Plxna1* and *Plxnb2* most highly expressed, followed by *Plxnd1, Plxna3,* and *Plxna2* (*Figure 1—figure supplement 1*). To address the effect of endogenous PLXNs on HH signaling in NIH/3T3 cells, we generated two different *Plxna1⁻/⁻;Plxna2⁻/⁻* mouse embryonic fibroblast (MEF) lines from embryonic day (E) 14.5 mouse embryos (*Todaro and Green, 1963*). We then used quantitative, real-time polymerase chain reaction (RT-qPCR) to analyze HH target gene expression in fibroblasts treated with a SMO agonist (SAG; *Figure 1H and I*). In each experiment, we used BLOCK-iT fluorescent oligos to visually confirm transfection and compared each result to an internal BLOCK-iT transfected control (*Figure 1H and I*). Therefore, fold changes in expression are relative within each experiment and should not be compared across panels. Interestingly, both cell lines lacking *Plxna1* and *Plxna2* still respond to SAG activation of HH signaling, as measured by expression of the direct HH transcriptional targets, *Gli1* and *Ptch1* (*Figure 1H and I*). We hypothesized that this was likely due to the presence of other Plxn family members, which could compensate for the lack of PLXNA1 and PLXNA2.

To address the potential functional redundancy of other Plxn family members, we used siRNA reagents to reduce levels of *Plxnb2, Plxna3,* and *Plxnd1* in *Plxna1⁻/⁻;Plxna2⁻/⁻* cells. Strikingly, both cell lines treated with the *Plxn* siRNAs listed above responded significantly less to SAG activation of *Gli1* and *Ptch1* compared to BLOCK-iT controls (*Figure 1H and I*). The degree of reduction following *Plxn* depletion is similar to that observed with *Nrp* depletion using previously published siRNA reagents targeting *Nrp1* and *Nrp2* (*Hillman et al., 2011*; *Figure 1H and I*). Together, these data suggest that, like NRPs, PLXNs are required for HH signal transduction in NIH/3T3 fibroblasts.

## The PLXNA1 transmembrane and cytoplasmic domains are necessary for HH signal promotion

PLXNs are single-pass TM proteins containing an ECD that can interact with NRPs and SEMA ligands, a TM domain that mediates dimerization, and a CD through which PLXNs signal intracellularly (*Neufeld and Kessler, 2008*). While many HH regulators at the cell surface bind to HH ligands through their ECD (*Lee et al., 2001*; *Tenzen et al., 2006*; *Capurro et al., 2008*; *Chang et al., 2011*; *Izzi et al., 2011*; *Christ et al., 2012*; *Whalen et al., 2013*), NRP1 acts through its CD to regulate HH signaling (*Ge et al., 2015*; *Pinskey et al., 2017*). To investigate the mechanism of PLXN action in HH signaling, we first asked whether the PLXN CD is required for HH promotion. Interestingly, deleting the PLXNA1 TM and CD (PLXNA1^ΔTMCD^) or the CD alone (PLXNA1^ΔCD^) abrogates PXLNA1-mediated promotion of HH signaling in NIH/3T3 cells (*Figure 2A and B*). Western blot analyses confirmed PLXNA1, PLXNA1^ΔTMCD^, and PLXNA1^ΔCD^ expression and PLXNA1^ΔTMCD^ secretion (*Figure 2C*). Further, immunofluorescence staining for an extracellular MYC epitope under permeabilizing and non-permeabilizing conditions confirmed the cell surface localization of PLXNA1 and PLXNA1^ΔCD^ as well as the secretion

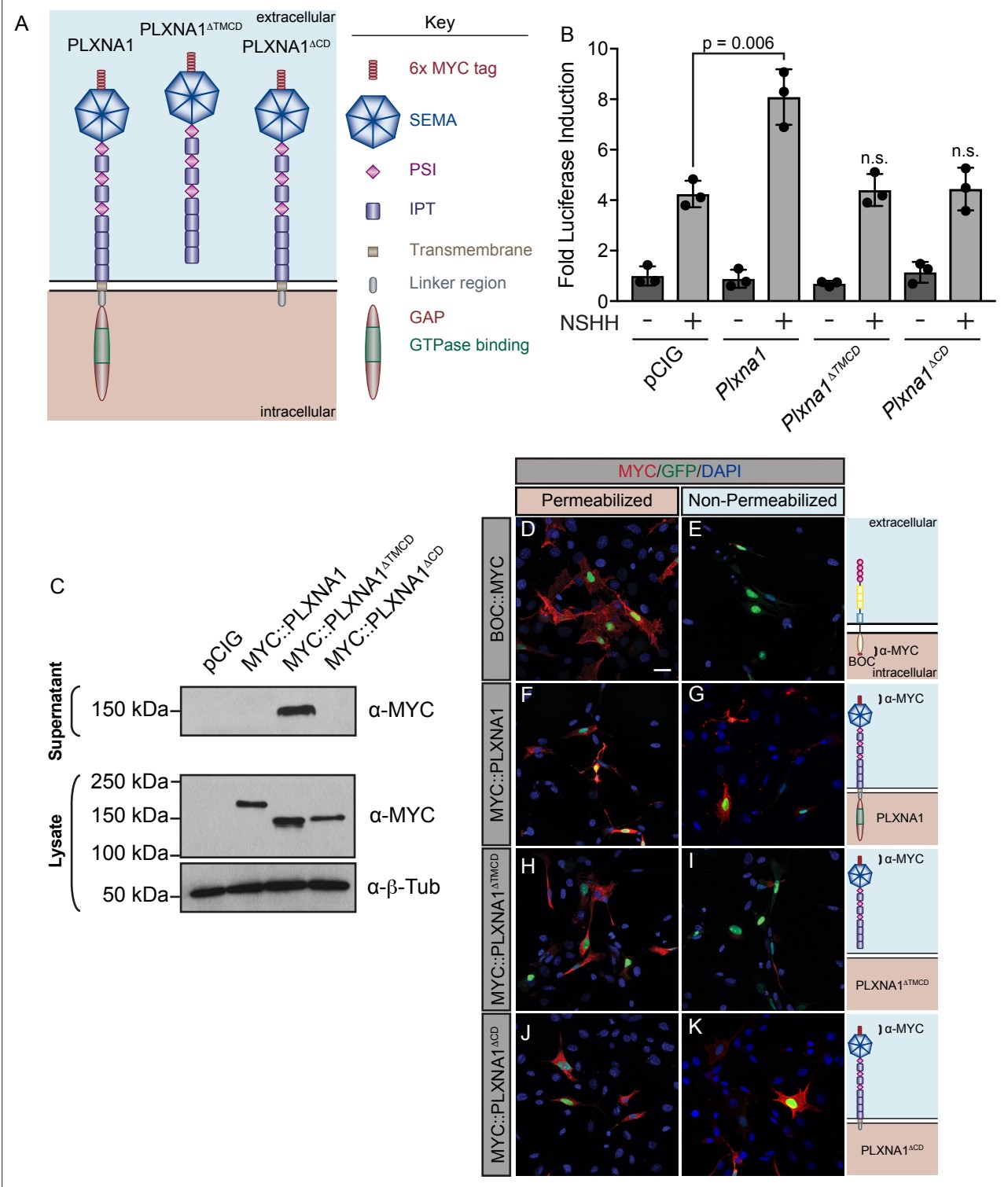

**Figure 2.** The PLXNA1 cytoplasmic and transmembrane domains are required for Hedgehog (HH) pathway promotion. (**A**) Schematic of different PLXNA1 proteins. (**B**) HH-dependent luciferase reporter activity was measured in NIH/3T3 cells transfected with the indicated constructs and stimulated with control (-NSHH) or NSHH-conditioned media (+NSHH). Data are reported as mean fold induction ± SD, with p-values calculated using two-tailed Student's *t*-tests. n.s., not significant. (**C**) Western blot analysis confirming expression of MYC-tagged PLXNA1 proteins in NIH/3T3 cells. Note that MYC::PLXNA1$^{\Delta TMCD}$ is detected in the supernatant, consistent with its predicted secretion. Anti-beta-tubulin (α-β-Tub) was used as a loading control. (**D–K**) Antibody detection of MYC (red) in permeabilized (left panels) and non-permeabilized (right panels) NIH/3T3 cells to assess cell surface localization of the indicated MYC-tagged proteins. Note that BOC, which contains a C-terminal MYC tag, is only detected under permeabilized conditions, while

*Figure 2 continued on next page*

Figure 2 continued

PLXNA1$^{\Delta TMCD}$, which is secreted, is also largely undetected under non-permeabilized conditions. Nuclear GFP (green) indicates transfected cells, whereas DAPI (blue) stains all nuclei. Diagrams (right) describe each construct, with brackets indicating antibody-binding sites. Scale bar = 10 μm.

The online version of this article includes the following source data for figure 2:

**Source data 1.** Raw data for *Figure 2B*.

**Source data 2.** Raw, unedited blot from *Figure 2C*.

**Source data 3.** Raw, unedited blot from *Figure 2C*.

**Source data 4.** Raw, unedited blot from *Figure 2C*.

**Source data 5.** Raw, labeled blot from *Figure 2C*.

**Source data 6.** Raw, labeled blot from *Figure 2C*.

**Source data 7.** Raw, labeled blot from *Figure 2C*.

of PLXNA1$^{\Delta TMCD}$ compared to a control BOC construct with a C-terminal MYC tag (*Figure 2D–K*). These results suggest that the PLXNA1 TM and CD are required for promotion of HH signaling.

## PLXN cytoplasmic GAP activity mediates HH signal promotion

Upon binding to the PLXN extracellular SEMA domain, SEMA ligand triggers a conformational change, releasing PLXN autoinhibition and allowing for the full activation of the intracellular GAP domain (*Takahashi and Strittmatter, 2001*; *Janssen et al., 2010*; *Nogi et al., 2010*). As a result, deleting the autoinhibitory PLXN ECD results in constitutive GAP activity that induces robust cytoskeletal collapse through downstream signaling events (*Takahashi and Strittmatter, 2001*; *Hota and Buck, 2012*). To further test whether PLXN GAP function regulates HH signaling, we deleted the PLXNA1 ECD (PLXNA1$^{\Delta ECD}$) and measured HH-dependent luciferase reporter activity in NIH/3T3 cells (*Figure 3A*). Not only is PLXNA1$^{\Delta ECD}$ still able to promote HH signaling, but the constitutively active PLXN GAP domain significantly augments the level of HH promotion (*Figure 3B*). While full-length PLXN boosts HH signaling one and a half- to twofold on average, PLXNA1$^{\Delta ECD}$ consistently increases the level of HH signaling between four- and tenfold, averaging an approximately sixfold increase (*Figure 3—figure supplement 1A*).

The PLXN CD is essential for intracellular SEMA signal transduction, acting through a split GAP domain to induce cytoskeletal collapse (*Püschel, 2007*; *Neufeld and Kessler, 2008*; *Duan et al., 2014*). Arginine to alanine mutations in residues 1429 and 1430 of mouse PLXNA1 disrupt GAP activity, rendering PLXNA1 a nonfunctional SEMA receptor in a COS7 cell collapse assay (*Rohm et al., 2000a*). Strikingly, recapitulating these conserved arginine mutations within the PLXNA1 GAP domain also rendered PLXNA1 unable to promote HH signaling (PLXNA1$^{R1}$; *Figure 3C*). Importantly, analogous mutations in PLXNB2 also abrogate the promotion of HH pathway activity (PLXNB2$^{R1}$; *Figure 3—figure supplement 1B and C*). Further, the A1R1 arginine to alanine GAP mutations in the context of the PLXNA1 ECD deletion significantly reduce the level of HH promotion, though they do not completely abrogate PLXN-mediated HH pathway induction when compared with PLXNA1$^{\Delta CD}$ (*Figure 3D*). To assess whether residual PLXN GAP activity contributes to this promotion, we mutated an additional conserved arginine residue (to alanine) in the carboxy terminal half of the split GAP domain (PLXNA1$^{R1R2\Delta ECD}$). However, this mutant still did not further abrogate the promotion of HH signaling when compared to PLXNA1$^{R1\Delta ECD}$ (*Figure 3—figure supplement 2B*). These data suggest that other cytoplasmic determinants contribute to PLXN-mediated HH promotion.

Previous work identified FYN kinase phosphorylation sites as key mediators of PLXN function (*St Clair et al., 2018*). While mutation of one tyrosine residue alone (PLXNA1$^{Y1\Delta ECD}$) or in conjunction with the A1R1 mutation (PLXNA1$^{R1Y1\Delta ECD}$) did not impact PLXN function (*Figure 3—figure supplement 2C*), mutation of a second tyrosine residue (PLXNA1$^{Y2\Delta ECD}$) abrogated promotion of HH signaling to comparable levels to the A1R1 mutation (*Figure 3—figure supplement 2C*). Notably, mutation of both the GAP domain and the second FYN kinase phosphorylation site (PLXNA1$^{R1Y2\Delta ECD}$) rendered PLXN completely inert (*Figure 3—figure supplement 2C*) in the context of HH signal transduction. Immunofluorescence analyses indicated appropriate localization of these constructs to the cell surface, compared to a C-terminally tagged BOC control, as well as cytoskeletal collapse in PLXNA1$^{\Delta ECD}$ and to some extent PLXNA1, with the expected lack of collapse in PLXNA1R1$^{\Delta ECD}$ and PLXNA1$^{\Delta CD}$

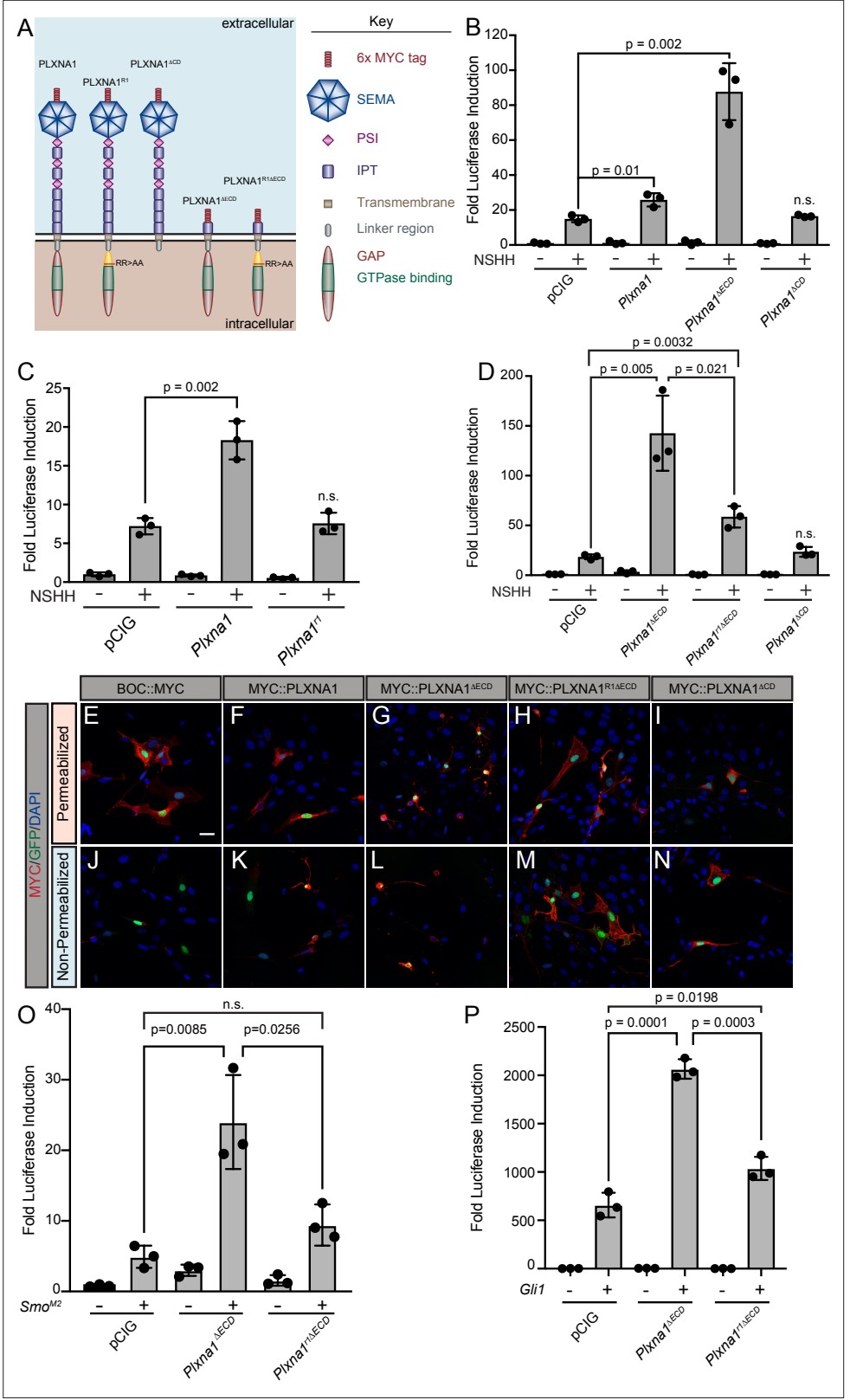

**Figure 3.** The plexin (PLXN) GTPase-activating protein (GAP) domain is required to promote Hedgehog (HH) signaling at the level of GLI transcription factors. (**A**) Schematic of different PLXNA1 proteins. (**B–D**) HH-dependent luciferase reporter activity was measured in NIH/3T3 cells transfected with the indicated constructs and stimulated with control (-NSHH) or NSHH-conditioned media (+NSHH). Data are reported as mean fold induction ± SD, with

*Figure 3 continued on next page*

*Figure 3 continued*

p-values calculated using two-tailed Student's *t*-tests. n.s., not significant. (**E–N**) Antibody detection of MYC-tagged proteins (red) in permeabilized (top panels) and non-permeabilized (bottom panels) NIH/3T3 cells to assess cell surface localization of the indicated constructs. Nuclear GFP (green) indicates transfected cells, whereas DAPI (blue) stains all nuclei. Note that constitutive PLXN GAP activity leads to cell collapse, as is observed with PLXNA1$^{\Delta ECD}$ and, to some extent, PLXNA1. For PLXNA1R1 localization, please see *Figure 3—figure supplement 1D and E*. Scale bar = 10 µm. (**O, P**) HH-dependent luciferase reporter activity was measured in NIH/3T3 cells transfected with the indicated constructs and stimulated by co-transfecting cells with pCIG, *Smo$^{M2}$* (**O**), or *Gli1* (**P**). Data are reported as mean fold induction ± SD, with p-values calculated using two-tailed Student's *t*-tests. n.s., not significant.

The online version of this article includes the following source data and figure supplement(s) for figure 3:

**Source data 1.** Raw data for *Figure 3B–D* and *Figure 3O and P*.

**Figure supplement 1.** Constitutively active PLXNA1 reproducibly increases Hedgehog (HH) pathway activity.

**Figure supplement 1—source data 1.** Raw data for *Figure 3—figure supplement 1C*.

**Figure supplement 2.** The plexin (PLXN) GTPase-activating protein (GAP) domain and a FYN kinase phosphorylation site contribute to PLXN-mediated promotion of Hedgehog (HH) signaling.

**Figure supplement 2—source data 1.** Raw data for *Figure 3—figure supplement 2B*.

**Figure supplement 3.** Plexins (PLXNs) inhibit WNT signaling.

**Figure supplement 3—source data 1.** Raw data for *Figure 3—figure supplement 3B and C*.

(*Figure 3E–N*, *Figure 3—figure supplement 1D–G*). Together, these results suggest that GAP activity and FYN kinase phosphorylation are necessary for PLXN-mediated promotion of HH signaling.

To examine whether PLXN-dependent promotion is specific to HH signaling or whether it has broader effects on additional signaling pathways, we again employed luciferase assays using a reporter construct containing multiple TCF/LEF binding sites (TOP-FLASH) to measure Wnt pathway activity (*Molenaar et al., 1996*; *Figure 3—figure supplement 3A*). Whereas either β-CATENIN expression or Chiron treatment significantly activates Wnt signaling in NIH/3T3 cells (*Figure 3—figure supplement 3B and C*), PLXNA1 does not promote Wnt pathway activation. Instead, PLXNA1, and to a greater degree PLXNA1$^{\Delta ECD}$, inhibit Wnt pathway activity, with PLXNA1$^{\Delta ECD}$ reducing Wnt pathway activity to baseline levels (*Figure 3—figure supplement 3B and C*). These data suggest that PLXN does not act to generally promote transcription, and instead has opposing consequences on HH and Wnt transcriptional readouts.

## PLXNA1 promotes HH signaling downstream of SMO

HH signaling culminates in the differential processing and activation of the GLI family of transcription factors, which shuttle in and out of the primary cilium and are phosphorylated by several kinases to regulate their activity (*Hui and Angers, 2011*). Transfecting *Smo$^{M2}$*, a constitutively active form of SMO, or *Gli1*, an obligate HH activator, into our luciferase reporter assay in NIH/3T3 cells results in tens to thousands of fold induction of HH reporter activity, respectively. Still, co-transfecting *Smo$^{M2}$* or *Gli1* with *Plxna1$^{\Delta ECD}$* results in a significantly greater HH response (*Figure 3O and P*). Notably, this promotion requires GAP activity as co-transfection of *Smo$^{M2}$* or *Gli1* with a GAP-deficient *Plxn* (*Plxna1$^{r1\Delta ECD}$*) returns HH pathway activation to near-baseline levels (*Figure 3O and P*). These data suggest that PLXNs function downstream of HH ligand at the level of GLI or transcriptional regulation, and that full PLXN GAP activation via the release of extracellular autoinhibition is necessary for enhanced HH promotion beyond the level observed with either *Smo$^{M2}$* or *Gli1* alone.

## PLXNs are not enriched in the primary cilium, but do require primary cilia for HH pathway promotion

The primary cilium is an important platform for HH signaling molecules (*Wong et al., 2009*; *Goetz and Anderson, 2010*) and many HH pathway components, including NRP, are enriched there (*Corbit et al., 2005*; *Haycraft et al., 2005*; *Rohatgi et al., 2007*; *Pinskey et al., 2017*). Notably, molecules over 40 kDa are unable to freely diffuse into the primary cilium, requiring active transport to enter this highly regulated subcellular compartment (*Kee et al., 2012*). To test whether PLXNs localize to the primary cilium, we expressed MYC-tagged PLXNs in NIH/3T3 cells and performed immunofluorescent

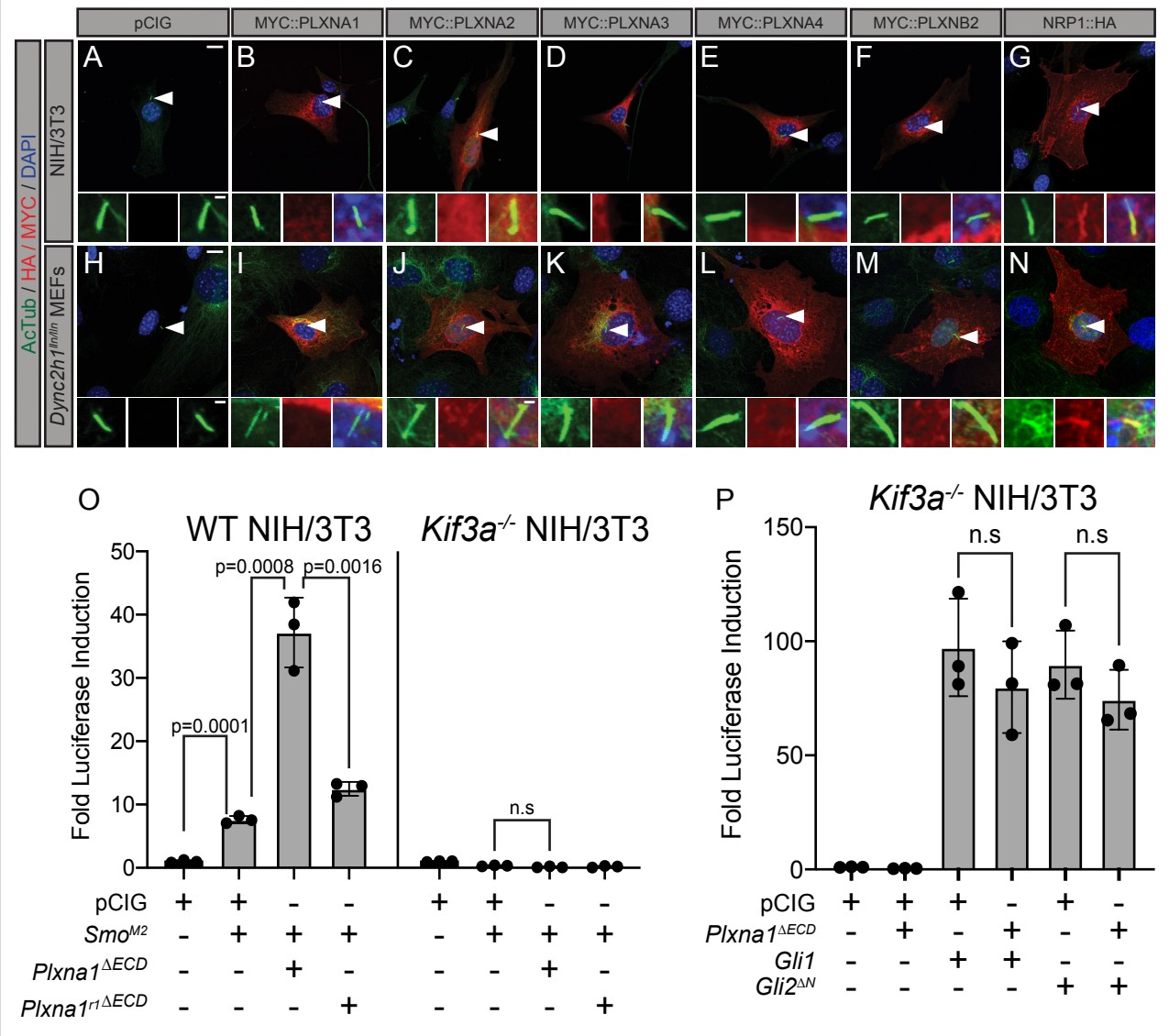

**Figure 4.** Plexins (PLXNs) do not localize to primary cilia, but do require primary cilia to promote Hedgehog (HH) pathway activity. (**A–N**) Antibody detection of MYC and HA-tagged constructs (red) in NIH/3T3 cells (**A–G**) and *Dync2h1$^{lln/lln}$* mouse embryonic fibroblasts (MEFs) (**H–N**). Acetylated tubulin (AcTub, green) indicates the primary cilium and DAPI (blue) stains nuclei. Compared to NRP1, PLXNs are not enriched in primary cilia. Scale bar = 10 μm. Inset scale bar = 1 μm. (**O**) WT NIH/3T3 cells or *Kif3a$^{-/-}$* NIH/3T3 cells were co-transfected with *Smo$^{M2}$* and *Plxna1$^{ΔECD}$* or *Plxna1r1$^{ΔECD}$*. (**P**) *Kif3a$^{-/-}$* NIH/3T3 were transfected with *Gli1* or *Gli2$^{ΔN}$* and co-transfected with *Plxna1$^{ΔECD}$*. Data are reported as mean fold induction ± SD, with p-values calculated using two-tailed Student's *t*-tests. n.s., not significant.

The online version of this article includes the following source data and figure supplement(s) for figure 4:

**Source data 1.** Raw data for *Figure 4O and P*.

**Figure supplement 1.** Plexins (PLXNs) do not affect Smoothened (SMO) accumulation in primary cilia, ciliary length, or ciliation frequency.

**Figure supplement 1—source data 1.** Raw data for *Figure 4—figure supplement 1E–I*.

staining for MYC and acetylated tubulin (AcTub), which marks the primary cilium. PLXNs are broadly localized throughout the cell (*Figure 4A–N*), including the cell surface (*Figure 3E–N*), but they are largely excluded from the nucleus. Unlike NRP1, PLXN staining was not enriched within the primary cilium for any of the constructs we tested (*Figure 4A–G*). MEFs with a mutation in the dynein heavy chain (*Dync2h1$^{lln/lln}$*) exhibit impaired retrograde transport within the cilium, allowing for more robust detection of accumulated proteins (*Ocbina et al., 2011*). However, even in *Dync2h1$^{lln/lln}$* MEFs, PLXNs

still do not accumulate in the primary cilium (*Figure 4H–N*). These data suggest that PLXN localization to primary cilia is not required to regulate HH signal transduction.

To examine a potential requirement for primary cilia in PLXN-dependent promotion of HH signaling, we performed luciferase assays in WT NIH/3T3 cells as well as $Kif3a^{-/-}$ NIH/3T3 cells, which fail to assemble primary cilia (*Engelke et al., 2019*). As expected, WT NIH/3T3 cells activate HH signaling in response to $Smo^{M2}$ transfection, while $Kif3a^{-/-}$ NIH/3T3 cells do not (*Figure 4O*). Notably, $Kif3a^{-/-}$ NIH/3T3 cells also do not respond to co-transfection with $Smo^{M2}$ and $Plxna1^{\Delta ECD}$ (*Figure 4O*). Both GLI1 and GLI2$^{\Delta N}$ have been reported to promote HH pathway activation in the absence of primary cilia (*Haycraft et al., 2005*; *Wong et al., 2009*). We confirmed these data by transfecting $Kif3a^{-/-}$ NIH/3T3 cells with either *Gli1* or $Gli2^{\Delta N}$ (*Figure 4P*). Strikingly, and distinct from what we observe in WT NIH/3T3 cells, co-transfecting $Kif3a^{-/-}$ NIH/3T3 cells with either *Gli1* or $Gli2^{\Delta N}$ and $Plxna1^{\Delta ECD}$ displayed no further promotion of HH signaling (*Figure 4P*; *Figure 3P*). These data suggest that, while PLXNs do not localize to the primary cilium, primary cilia are required for PLXN-dependent promotion of HH signaling. Further, *Plxna1* transfection does not affect SMO localization to primary cilia, ciliary length, or the rate of ciliation amongst cells after treatment with HH-conditioned media or control-conditioned media (*Figure 4—figure supplement 1A–G*). Importantly, no difference was observed in vivo in ciliary length or ciliation frequency in the dentate gyrus of $Plxna2^{-/-}$ mice compared to heterozygous littermates (*Figure 4—figure supplement 1H and I*).

## Constitutive Plxn GAP activity drives ectopic cell migration in the embryonic chicken neural tube

The developing spinal cord requires HH signaling for proper patterning and development (*Dessaud et al., 2008*). SHH, which is initially secreted from the notochord, signals in a ventral–dorsal gradient to specify distinct cell fates in the neural tube. Notably, SHH also controls cell proliferation and cell migration in this tissue (*Cayuso and Martí, 2005*; *Cayuso et al., 2006*). Previous work demonstrated that multiple *Plxns* are expressed in the developing chicken neural tube concomitant with SHH-dependent tissue patterning (*Mauti et al., 2006*). To investigate potential contributions of PLXNs to these SHH-dependent outcomes, we employed chicken in ovo neural tube electroporation. While electroporation with an empty vector (pCIG) does not impact neural tube patterning (*Figure 5A–D*), $Smo^{M2}$ electroporation drives ectopic expression of NKX6.1, a direct target of HH signaling that is normally restricted ventrally, in the dorsal neural tube (*Figure 5E–H*). Similarly, electroporation with *Gli1*, an obligate activator of the HH pathway that drives high levels of HH signaling, also results in expansion of the NKX6.1 domain (*Figure 5I–L*). In some $Plxna1^{\Delta ECD}$-electroporated embryos, we observed apparent ectopic NKX6.1 expression (*Figure 5P*, *Figure 5—figure supplement 1K*, yellow arrowheads); however, quantitation of the NKX6.1 domain size revealed no significant differences between pCIG- and $Plxna1^{\Delta ECD}$-electroporated embryos (*Figure 5—figure supplement 1J–M*). Further, cells electroporated with $Plxna1^{\Delta ECD}$ at the periphery of the endogenous NKX6.1 domain do not express NKX6.1, while cells in this same region that were electroporated with *Gli1* are NKX6.1 positive (*Figure 5—figure supplement 1J–L*).

*Gli1* expression also results in ectopic migration of cells into the dorsal lumen of the neural tube, which is typically completely devoid of cells (*Figure 5I*, yellow asterisk). Electroporation of $Plxna1^{\Delta ECD}$ phenocopies *Gli1*-induced migration into the lumen of the neural tube (*Figure 5M–P*, yellow asterisk), with a similar loss of PAX7-positive cells on the electroporated side (*Figure 5—figure supplement 1A–I*). Importantly, PLXN-dependent ectopic cell migration is lost upon mutation of the intracellular PLXN GAP domain (*Figure 5Q–T*, white asterisk). To analyze whether the PLXN-mediated ectopic cell migration is HH-dependent, we co-electroporated neural tubes with $Plxna1^{\Delta ECD}$ and $Ptch1^{\Delta L2}$, a constitutively active form of *Ptch1* that is insensitive to HH ligands (*Briscoe et al., 2001*). Consistent with previous reports, $Ptch1^{\Delta L2}$ expression inhibits endogenous HH pathway activity, visualized by the cell-autonomous loss of NKX6.1 in electroporated cells in the ventral neural tube (*Figure 6F–J*). Whereas cell migration is observed in the dorsal lumen of $Plxna1^{\Delta ECD}$-electroporated neural tubes (*Figure 6K–O*, yellow asterisk), cells co-electroporated with $Ptch1^{\Delta L2}$ and $Plxna1^{\Delta ECD}$ no longer migrate (*Figure 6P–T*). These data suggest that increased PLXN-mediated migration in the neural tube is HH-dependent.

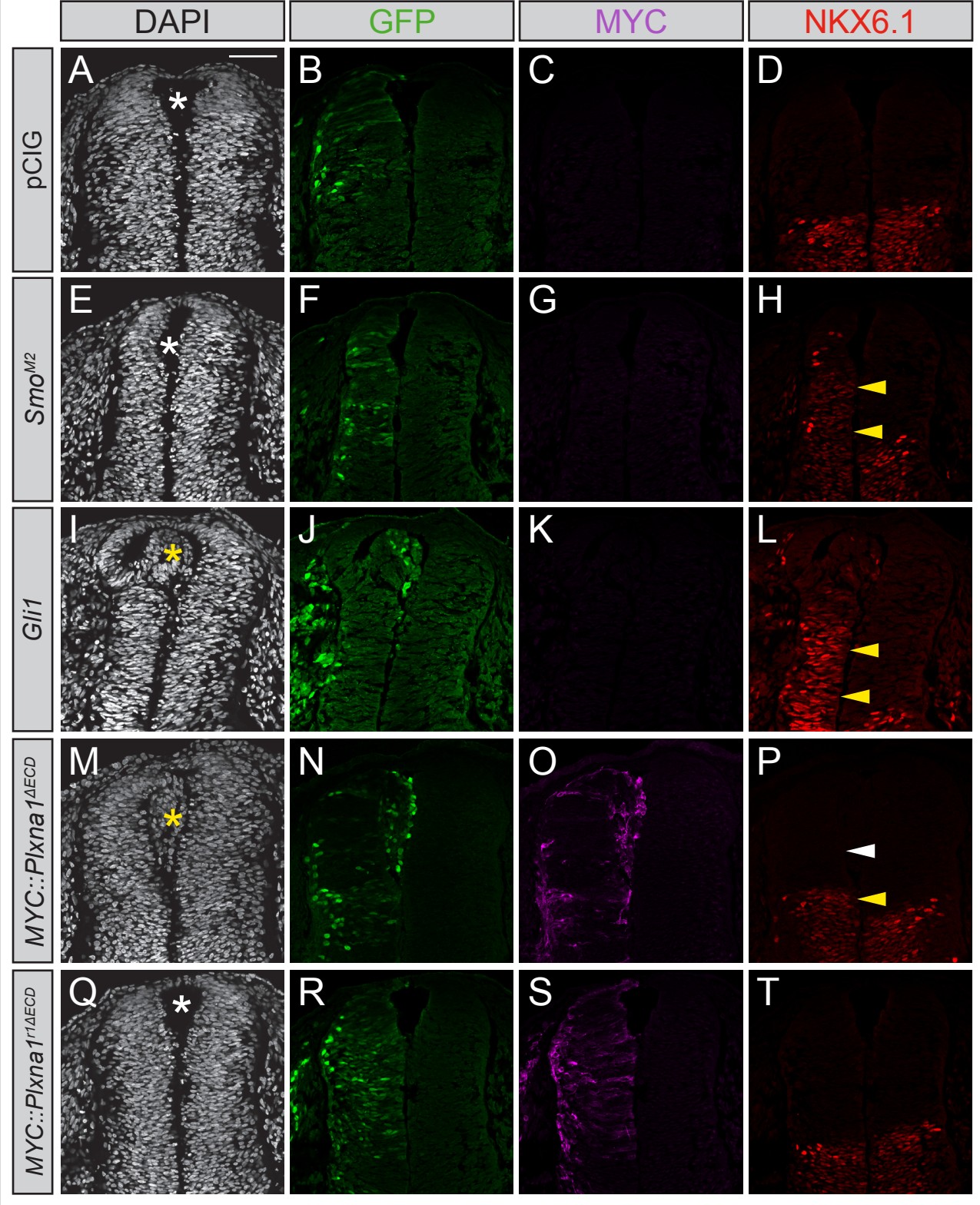

**Figure 5.** Constitutively active PLXNA1 induces ectopic cell migration into the lumen of the developing chicken neural tube. (**A–T**) Immunofluorescent analysis of neural patterning in forelimb-level sections from Hamburger–Hamilton stage 21–22 chicken embryos. Embryos were electroporated at Hamburger–Hamilton stage 11–13 with pCIG (**A–D**, n = 6 embryos), $Smo^{M2}$ (**E–H**, n = 7 embryos), $Gli1$ (**I–L**, n = 4 embryos), $MYC::Plxna1^{\Delta ECD}$ (**M–P**, n = 17 embryos), or $MYC::Plxna1r1^{\Delta ECD}$ (**Q–T**, n = 6 embryos). Transverse sections were stained with GFP, MYC, and NKX6.1 antibodies. DAPI stain labels

*Figure 5 continued on next page*

*Figure 5 continued*

nuclei (gray). Electroporated cells are labeled with GFP. Asterisks denote the presence (yellow) or absence (white) of ectopic cells within the lumen of the neural tube. Arrowheads denote the presence (yellow) or absence (white) of ectopic NKX6.1. Scale bar = 50 μm.

The online version of this article includes the following source data and figure supplement(s) for figure 5:

**Figure supplement 1.** Constitutively active PLXNA1 does not significantly alter Hedgehog-dependent neural tube patterning in the developing chicken embryo.

**Figure supplement 1—source data 1.** Raw data for *Figure 5—figure supplement 1M*.

## Plxna1 or Plxna2 deletion results in decreased numbers of HH-responding cells within the dentate gyrus

*Plxns* are expressed widely throughout the developing mouse embryo, particularly in the central nervous system (*Perälä et al., 2005*). Interestingly, developing neurons and progenitor cells in the hippocampus express *Plxns* (*Cheng et al., 2001*) and neuronal progenitor cells rely on HH signaling for proliferation and maintenance, particularly within the dentate gyrus (*Machold et al., 2003*; *Ahn and Joyner, 2005*). To determine whether PLXNs impact HH signaling in the hippocampus, we crossed

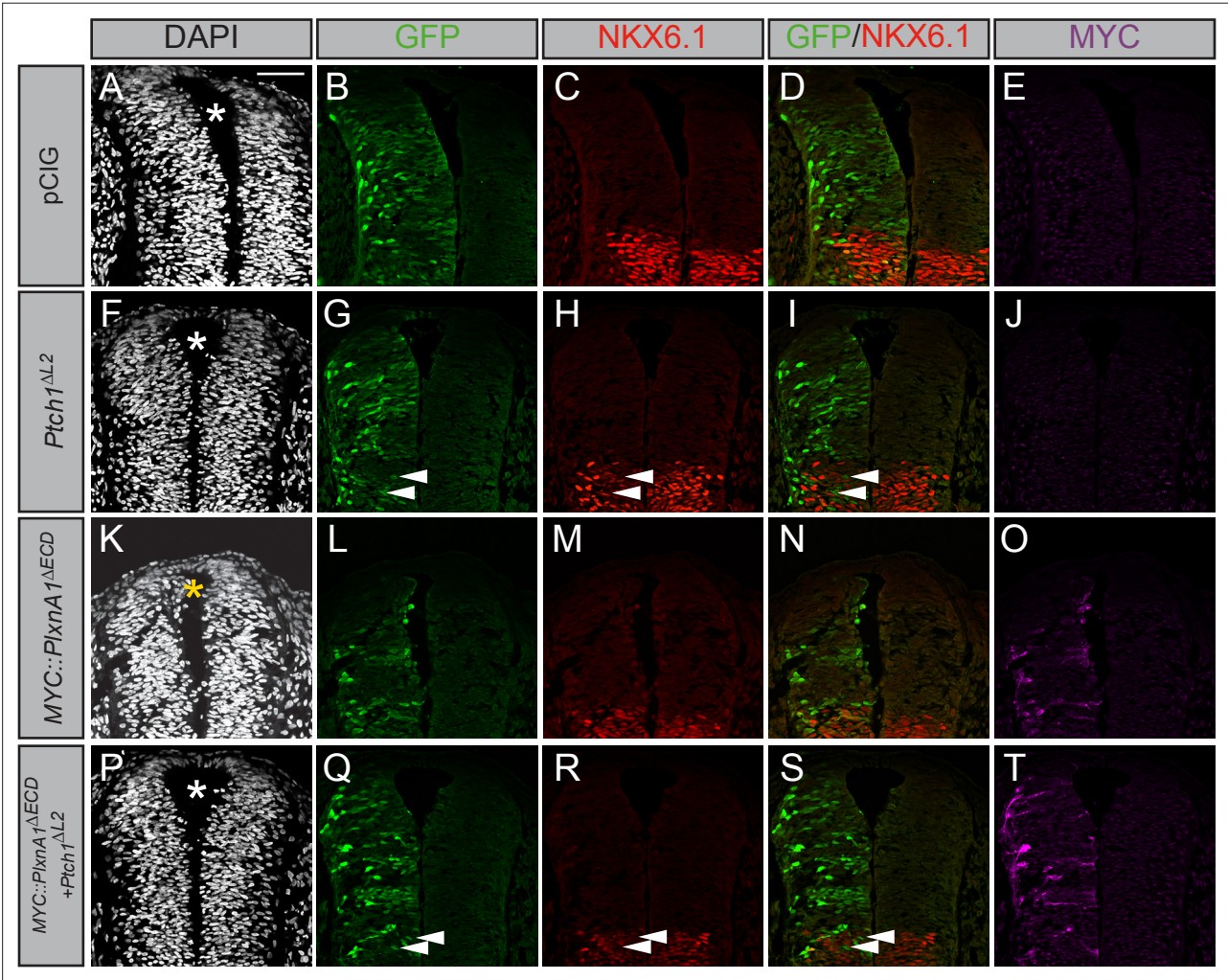

**Figure 6.** Plexin (PLXN)-mediated ectopic cell migration is Hedgehog (HH)-dependent. (**A–T**) Immunofluorescent analysis of neural patterning in forelimb-level sections from Hamburger–Hamilton stage 21–22 chicken embryos. Embryos were electroporated at Hamburger–Hamilton stage 11–13 with pCIG (**A–E**, n = 6 embryos), MYC::*Plxna1ᴬᴱᶜᴰ* (**F–J**, n = 7 embryos), *Ptch1ᴬᴸ²* (**K–O**, n = 5 embryos), or *MYC::Plxna1ᴬᴱᶜᴰ* and *Ptch1ᴬᴸ²* (**P–T**, n = 8 embryos). Transverse sections were stained with GFP, MYC, and NKX6.1 antibodies. DAPI stain labels nuclei (gray). Electroporated cells are labeled with GFP. Asterisks denote the presence (yellow) or absence (white) of ectopic cells within the lumen of the neural tube. Arrowheads denote absence of NKX6.1 in electroporated cells. Scale bar = 50 μm.

*Plxna1*- and *Plxna2*-deficient mice with a HH-responsive *Gli1^lacZ* reporter allele and examined β-galactosidase activity along the rostroventral axis of the dentate gyrus at postnatal day 7 (P7). *Plxna1^-/-* mice have significantly fewer *Gli1*-positive cells in both the dorsal and ventral dentate gyrus compared to their heterozygous littermates (*Figure 7A–F*). *Plxna2* deletion has a similar effect on *Gli1* expression with significantly fewer β-galactosidase-positive cells detected in the hilus and subgranular zone of the dorsal and ventral dentate gyrus (*Figure 7G–L*). Notably, these phenotypes are similar to previously reported HH loss-of-function studies (*Machold et al., 2003*). Further, no differences were observed in BrdU+ or TUNEL+ cells in *Plxna1^-/-* or *Plxna2^-/-* mice compared to heterozygous littermates, indicating that *Plxna1* or *Plxna2* deletion does not significantly impact cell proliferation or cell death, respectively (*Figure 7—figure supplement 1A–L*, *Figure 7—figure supplement 2A–L*). Unfortunately, *Plxna1^-/-*; *Plxna2^-/-* embryos die prenatally, precluding analyses of any potential additive or synergistic effects on HH pathway activity in the postnatal dentate gyrus. Together, these data suggest that PLXNs regulate HH pathway activation in vivo and suggest that multiple PLXNs regulate HH signaling in the developing mouse hippocampus.

## Discussion

HH signaling plays important roles in tissue formation, homeostasis, and repair, coordinating many cellular processes, including proliferation, fate specification, and survival (*Briscoe and Thérond, 2013*). Canonical SEMA receptors, the NRPs and PLXNs, are expressed in a wide variety of tissues during active HH regulation (*Kawasaki et al., 1999*; *Perälä et al., 2005*; *Mauti et al., 2006*; *Perälä et al., 2012*). Here, we present evidence that PLXNs positively regulate HH signaling. Unlike many previously described cell surface HH regulators, which interact directly with HH ligands, PLXNs promote HH signaling through their CDs at the level of GLI regulation (*Figure 8*). More specifically, we find that GAP enzymatic activity within the PLXN CD is required for HH promotion, and that constitutive GAP activity further amplifies the HH response. This shows that the PLXN GAP domain is important for canonical SEMA signaling as well as amplification of HH signaling. Further, we find that, while PLXNs themselves do not localize to primary cilia, they require primary cilia to promote HH pathway activity. Finally, our data indicate that increased *Plxn* activity in ovo increases cell migration into the neural tube lumen, and *Plxn* deletion in vivo results in reduced HH pathway activity in mice. Taken together, we provide multiple lines of evidence for a novel role of PLXNs in HH pathway regulation.

### Semaphorin receptors act promiscuously in multiple signaling pathways

While NRPs and PLXNs were first described as SEMA receptors, they also function within other signaling pathways (*He and Tessier-Lavigne, 1997*; *Kolodkin et al., 1997*; *Takahashi et al., 1999*; *Tamagnone et al., 1999*). NRPs play roles in VEGF signaling to regulate angiogenesis, and they interact with a wide variety of proteins, including PIGF-2, heparan sulfate, TGF-β1, HGF, PDGF, FGF, L1-CAM, integrins, and SARS-CoV-2 spike protein (*Roth et al., 2008*; *Prud'homme and Glinka, 2012*; *Muhl et al., 2017*; *Sarabipour and Mac Gabhann, 2021*). PLXNs also form complexes with off-track, MET, Ron, scatter factor, Ig-CAMs, and VEGFR2 under various cellular conditions (*Winberg et al., 2001*; *Giordano et al., 2002*; *Conrotto et al., 2004*; *Toyofuku et al., 2004*). This raises many questions about the nature of these receptors' activities within individual and overlapping signaling contexts. For example, what factors determine whether PLXNs and NRPs function as SEMA receptors or whether they regulate HH signaling? Can these processes happen simultaneously, and if so, how do they influence one another?

Multiple lines of evidence link altered SEMA/PLXN signaling to cancer. Depending on context, aberrant SEMA signaling may promote or suppress tumor growth and lead to various types of cancer (*Neufeld et al., 2016*). The mechanisms by which altered PLXN signaling influence tumor growth are incompletely understood. A link to increased HH signaling is intriguing because of the well-established role of elevated HH signaling in malignancies.

Another outstanding question is how SEMA ligands impact HH signaling. The role of SEMA ligands in HH pathway promotion remains unclear as conflicting pieces of evidence exist in the literature. In one study, addition of SEMA ligands in combination with HH ligand or SAG increased HH signaling in NIH/3T3 cells (*Ge et al., 2015*). Conversely, blocking NRP interaction with SEMA ligand reduces GLI expression (*Ge et al., 2015*). This model suggests that SEMA ligand increases recruitment of PDE4D

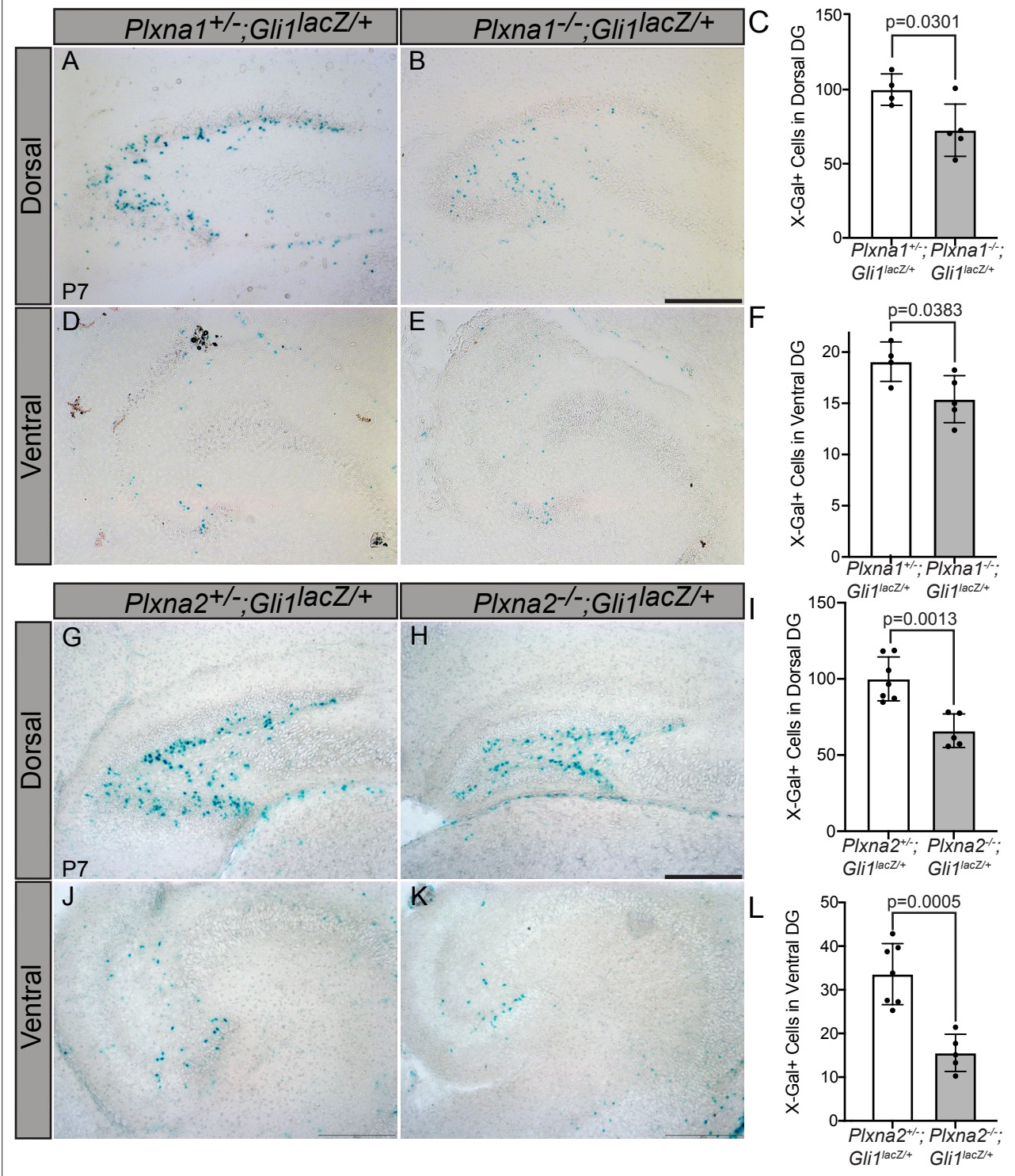

**Figure 7.** Reduced *Gli1^lacZ* expression in the dentate gyrus (DG) of mice lacking either *Plxna1* or *Plxna2*. X-Gal staining in coronal sections through the dorsal (**A, B, G, H**) and ventral (**D, E, J, K**) hippocampus of postnatal day 7 (P7) mice. The following numbers of pups were analyzed: *Plxna1^+/-;Gli1^lacZ/+* (n = 4); *Plxna1^-/-;Gli1^lacZ/+* (n = 5); *Plxna2^+/-;Gli1^lacZ/+* (n = 7); *Plxna2^-/-;Gli1^lacZ/+* (n = 5). Quantitation of *Gli1^lacZ*-positive cells (**C, F, I, L**) reported as mean ± SD, with p-values calculated using two-tailed Student's *t*-test. Scale bar = 200 μm.

The online version of this article includes the following source data and figure supplement(s) for figure 7:

**Source data 1.** Raw data for *Figure 7C, F, I and L*.

**Figure supplement 1.** *Plxna1* and *Plxna2* deletion do not alter cell proliferation in the hippocampus.

*Figure 7 continued on next page*

*Figure 7 continued*
**Figure supplement 1—source data 1.** Raw data for *Figure 7—figure supplement 1C, F, I and L*.
**Figure supplement 2.** *Plxna1* and *Plxna2* deletion does not alter apoptosis in the hippocampus.
**Figure supplement 2—source data 1.** Raw data for *Figure 7—figure supplement 2C, F, I and L*.

to the cell membrane, which interacts with the NRP CD and inhibits PKA, a negative regulator of GLI proteins (*Ge et al., 2015*). However, other studies suggest that addition of SEMA ligand has no effect on HH signaling (*Hillman et al., 2011*), and that NRPs still promote HH signaling when co-transfected with a version of GLI2 that cannot be phosphorylated by PKA at seven important sites (*Pinskey et al., 2017*). It is important to consider that NIH/3T3 cells, in which these studies were performed, express endogenous PLXNs (*Figure 1—figure supplement 1*). Given the results presented here, an alternate explanation of SEMA-mediated HH promotion is that SEMA ligands act through endogenous PLXNs to increase HH reporter activity by stimulating GAP activity. It is also possible that PLXNs themselves or PLXN-NRP complexes recruit PDE4D to the cell membrane rather than NRPs alone. Another discrepancy in the literature concerns the requirement for the NRP ECD in HH promotion (*Ge et al., 2015*; *Pinskey et al., 2017*). Again, given that PLXNs promote HH signaling and that the NRP ECD mediates interactions with PLXN co-receptors, the variable effects that have been reported could be explained by the presence of endogenous PLXNs, the level of NRP overexpression, and the sensitivity of the assay. Future studies should investigate the effects of PLXN-mediated HH promotion in the absence of NRPs and vice versa to further elucidate their mechanisms of action.

## NRP and PLXN cooperation in HH signaling

We previously reported that NRPs promote HH signaling through a novel cytoplasmic motif (*Pinskey et al., 2017*), within a region of the protein that is dispensable for SEMA signaling (*Fantin et al., 2011*). This suggests that NRPs may act very differently within SEMA and HH signaling contexts. PLXNs, on the other hand, seem to function similarly in HH and SEMA signaling through cytoplasmic GAP activity. Together, these data raise the question: do NRPs and PLXNs function together or separately in HH signaling? The answer may be both. Several pieces of evidence suggest that NRPs

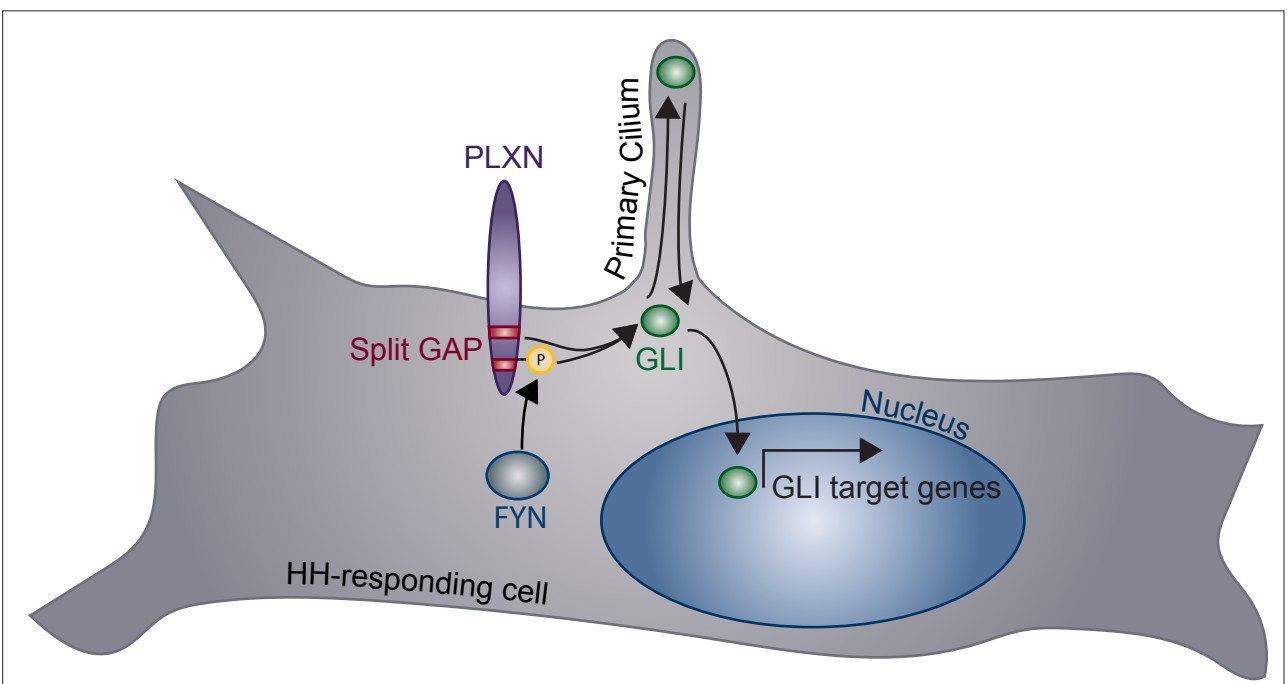

**Figure 8.** Model of plexin (PLXN)-mediated promotion of Hedgehog (HH) pathway activity. PLXNs (purple) at the cell surface promote HH signaling through GLI transcription factor (green) activation, mediated by their cytoplasmic GTPase-activating protein (GAP) activity (red) and FYN kinase phosphorylation (yellow). Notably, this PLXN-dependent promotion requires primary cilia to induce GLI target gene expression in the nucleus.

function independently of PLXNs in HH signaling. First, deleting the NRP ECD, which mediates interaction between NRPs, PLXNs, and SEMA ligands, does not disrupt HH pathway promotion (*Pinskey et al., 2017*). Furthermore, we report here that PLXNB2 can promote HH signaling, despite its lack of reported interactions with NRPs (*Neufeld and Kessler, 2008*). However, we cannot exclude the possibility that PLXN A subfamily members bind to endogenous NRPs to mediate HH promotion in our assays. Therefore, the ideas that NRPs and PLXNs function independently and together in HH signaling are not mutually exclusive, and additional studies will be required to elucidate their independent and/or cooperative roles.

### Connecting PLXN GAP activity to the HH pathway

We find that HH pathway activity is regulated by enzymatic activity of the PLXN GAP domain. However, it remains unclear how GAP downstream signaling intersects with the HH signal cascade. The PLXN CD interacts with a plethora of intracellular proteins, including collapse-response-mediator protein (CRMP) family phosphoproteins, protein kinases, MICAL redox proteins, and small intracellular GTPases from the Rho, Ras, and Rap superfamilies (*Püschel, 2007*; *Yang and Terman, 2013*; *Jongbloets and Pasterkamp, 2014*). Further, our understanding of the cellular mechanisms downstream of the PLXN GAP domain remains incomplete, including which GTPases are regulated by various PLXN family members. This makes it difficult to identify candidates that might mediate HH signaling. Here, we find that PLXNs from both the A and B subfamilies can promote HH signaling, which may be an important clue in answering this question. While we cannot exclude the possibility that each PLXN or PLXN subfamily regulates HH differently, it is likely that they converge upon a common protein or set of proteins that mediate HH promotion. Our data suggest that this convergence takes place at the level of GLI transcription factors and requires intact primary cilia. Therefore, candidates for future study should have common demonstrated roles downstream of all PLXNs.

### PLXN redundancy in HH pathway promotion

As previously discussed, the PLXN family of proteins is comprised of nine members with distinct and overlapping functions (*Neufeld and Kessler, 2008*). One shared feature between all PLXN proteins is the conserved cytoplasmic GAP domain (*Neufeld and Kessler, 2008*), which we find mediates HH signal promotion. Therefore, our results are complicated by the presence of endogenous PLXN proteins that may act redundantly in the HH signaling cascade, particularly given that PLXNs from multiple subfamilies promote HH signaling. Though technically challenging, a PLXN null background would be necessary to truly study the combined function of PLXN family members in HH signaling. It is also important to consider that PLXNs exhibit largely overlapping expression patterns in vivo, further complicating loss-of-function studies (*Perälä et al., 2005*; *Mauti et al., 2006*). Notably, our results suggest that deleting *Plxna1* or *Plxna2* alone is sufficient to reduce HH target gene expression in the dentate gyrus (*Figure 7*), despite the widespread expression of additional *Plxns* in the central nervous system (*Perälä et al., 2005*), including *Plxna3*, which is highly expressed in the developing hippocampus (*Cheng et al., 2001*). Our current study is limited to analysis of individual *Plxn* mutant animals– future work investigating the consequences of combined *Plxn* deletion will provide greater insight into PLXN regulation of HH pathway activity.Additional HH-responsive tissues that express a smaller subset of *Plxns*, including the olfactory epithelium, the tooth bud, and the lung (*Perälä et al., 2005*), should be considered for broader in vivo studies.

Our study and many others highlight the complex, entangled nature of cell signaling molecules and pathways. While they are typically studied in isolation, it may be useful to instead consider signaling pathways as broader signaling networks, with overlapping inputs and outputs that combine to elicit cellular behaviors. By better understanding these systems, we can begin to decode the factors influencing cellular decision-making in developmental, homeostatic, and diseased states.

## Materials and methods
### Plxn constructs

*Plxn* constructs were derived from full-length cDNAs using standard molecular biology techniques. All constructs were cloned into the pCIG vector, which contains a CMV enhancer, a chicken beta-actin promoter, and an internal ribosome entry site (IRES) with a nuclear enhanced green fluorescent

protein reporter (3XNLS-EGFP) (*Megason and McMahon, 2002*). C-terminal or N-terminal 6X MYC tags (EQKLISEEDL) were added to constructs as indicated. Deletion and mutation variants were generated using standard cloning techniques and the QuikChange II XL Site-Directed Mutagenesis Kit (Agilent Technologies, 200521).

## Cell culture and MEF generation

Cell lines were maintained in Dulbecco's Modified Eagle Medium (DMEM; Thermo Fisher Scientific, 11965-118) supplemented with 10% bovine calf serum (ATCC, 30-2030) and 1X Penicillin–Streptomycin–Glutamine (Life Technologies, 10378016). Cultures were maintained at 37°C with 5% $CO_2$ and 95% humidity. MEFs were generated as previously described (*Todaro and Green, 1963*). NIH/3T3 cells (CRL-1658) and COS-7 cells (CRL-1651) were purchased from ATCC (Cat# CRL-1658). *Plxna1$^{-/-}$;Plxna2$^{-/-}$* MEFs were generated in the laboratory and authenticated using PCR. All cell lines were mycoplasma negative. *Kif3a-/-* NIH/3T3 Flp-In cells were obtained from Dr. Kristen Verhey (*Engelke et al., 2019*).

## Cell signaling assays

Luciferase-based reporter assays in NIH/3T3 cells were performed as previously described using a ptcΔ136-GL3 reporter construct to measure HH activity (*Nybakken et al., 2005*) or TOP-FLASH for Wnt activity (*Molenaar et al., 1996*). Briefly, cells were seeded at $2.5 \times 10^4$ cells/well into 0.5% gelatin-coated 24-well plates. The next day, cells were transfected with empty vector (pCIG) or experimental constructs along with the *ptcΔ136-GL3* luciferase reporter construct and beta-galactosidase transfection control (pSV-β-galactosidase; Promega, E1081). Transfections were performed using Lipofectamine 2000 (Invitrogen, 11668) and Opti-MEM reduced serum media (Invitrogen, 31985). Then, 48 hr after transfection, culture media were replaced with low-serum media (0.5% bovine calf serum, 1% Penicillin–Streptomycin L-glutamine) containing either control, N-terminal SHH (NSHH)-conditioned media, DMSO, 300 nM SAG (Enzo Life Sciences, ALX-270-426-M001), or 30 µM Chiron (APExBIO, A3011). Luciferase reporter activity and beta-galactosidase activity were measured 48 hr later on a Spectramax M5$^e$ Plate reader (Molecular Devices) using the Luciferase Assay System (Promega, E1501) and the Betafluor Beta Galactosidase Assay Kit (EMD Millipore, 70979), respectively. Luciferase values were divided by beta-galactosidase activity to control for transfection, and data were reported as fold induction relative to the vector-transfected control. All treatments were performed in triplicate (each data point indicates a technical replicate) and averaged (bar height), with error bars representing the standard deviation between triplicate wells. Each experiment was repeated a minimum of three times (biological replicates); representative results are shown. Student's *t*-tests were used to determine whether each treatment was significantly different from the control, with p-values of 0.05 or less considered statistically significant.

## Immunofluorescent analyses for cultured cells

NIH/3T3 fibroblasts were plated at $1.5 \times 10^5$ cells/well onto glass coverslips in a 6-well dish. Cells were transfected 24 hr after plating using Lipofectamine 2000 (Invitrogen, 11668) and Opti-MEM reduced serum media (Invitrogen, 31985). To assess expression and collapse, cells were incubated for 24–48 hr at 37°C as indicated. To image cilia, cells were placed in low-serum media approximately 6 hr after transfection (0.5% bovine calf serum, 1% Penicillin–Streptomycin L-glutamine) for 48 hr. All cells were fixed in 4% paraformaldehyde for 10 min at room temperature and washed with PBS. A 5 min permeabilization step with 0.2% Triton X-100 in PBS was performed as indicated, prior to staining. Primary antibodies included mouse IgG2a anti-MYC (1:1000, Cell Signaling, 2276), goat IgG anti-PLXNA1 (1:250, R&D Systems, AF4309), mouse IgG2b anti-acetylated tubulin (1:2500, Sigma-Aldrich, T7451), rabbit IgG anti-Arl13B (1:2500, Proteintech, 17711-1-AP), or mouse IgG2a anti-Smoothened (1:50, Santa Cruz Biotechnology, sc-166685), all diluted in IF blocking buffer (30 g/L bovine serum albumin, 1% heat-inactivated sheep serum, 0.02% $NaN_3$, and 0.1% Triton X-100 in PBS). Coverslips were incubated with primary antibodies overnight, followed by a 10 min DAPI stain (1:30,000 in PBS at room temperature, Invitrogen, D1306) and 1 hr incubation with secondary antibodies including Alexa Fluor 555 goat anti-mouse IgG2a, Alexa Fluor 488 donkey anti-goat IgG, Alexa Fluor 488 goat anti-mouse IgG2b, and Alexa Fluor 555 goat anti-mouse IgG2b (1:500, Invitrogen, A21137, A11055, A21141, and A21147, respectively). Coverslips were mounted

to glass slides using Shandon Immu-Mount Mounting Medium (Fisher, 9990412). Immunofluorescent analyses and imaging were performed on a Leica SP5X Upright 2-Photon Confocal microscope using LAS AF software (Leica) and a Leica 63× (type: HC Plan Apochromat CS2; NA1.2) water immersion objective. Cilia length was measured using ImageJ. Ciliary SMO signal, as measured by overlay with ARL13B, was quantified using ImageJ, and signal intensity was normalized to background as assessed by quantitation of an adjacent acellular area. Average values are represented by bar height with error bars representing standard deviation among samples. Student's $t$-tests were used to determine whether each treatment was significantly different from the control, with p-values of 0.05 or less considered statistically significant. To determine ciliation frequency, ciliated and non-ciliated cells were counted from randomly selected frames on stained slides. Total ciliated and non-ciliated cells were calculated from three replicates. Conditions were statistically compared using a chi-squared test.

## Western blot analysis

NIH/3T3 cells were transfected using Lipofectamine 2000 (Invitrogen, 11668) and Opti-MEM reduced serum media (Invitrogen, 31985). Cells were lysed in radioimmunoprecipitation assay (RIPA) buffer (50 mM Tris–HCl, pH 7.2, 150 mM NaCl, 0.1% Triton X-100, 1% sodium deoxycholate, and 5 mM EDTA) 48 hr after transfection, sonicated using a Fisher Scientific Sonic Dismembrator, Model 500 (four pulses at 20%), and centrifuged at 14,000 × $g$ for 25 min at 4°C to remove the insoluble fraction. Protein concentrations were determined using the BCA Protein Assay Kit (Fisher, PI23225). After boiling for 10 min, 50 µg of protein from each sample were separated using SDS-PAGE with 7.5–12.5% gels and transferred onto Immun-Blot PVDF membranes (Bio-Rad, 162-0177). Membranes were washed in Tris-buffered saline (TBS) with 0.5% OmniPur Tween-20 (TBST; EMD Millipore, 9480) and blocked in Western blocking buffer (30 g/L bovine serum albumin with 0.2% NaN$_3$ in TBST) for 1 hr to overnight. Blots were probed with the following antibodies: rabbit IgG anti-MYC (1:10,000, Bethyl Labs, A190-105A), goat IgG anti-PLXNA1 (1:200, R&D Systems, AF4309), and mouse IgG1 anti-beta tubulin (1:10,000, generously provided by Dr. Kristen J. Verhey, University of Michigan). Secondary antibodies from Jackson ImmunoResearch were diluted 1:10,000 and included peroxidase-conjugated AffiniPure goat anti-mouse IgG, light chain specific (115-035-174), peroxidase-conjugated AffiniPure F(ab)2 Fragment donkey anti-rabbit IgG (711-036-152), and peroxidase-conjugated AffiniPure donkey anti-goat IgG, light chain specific (705-035-147). Immobilon Western Chemiluminescent HRP Substrate (EMD Millipore, WBKLS0500) was added for 10 min before membranes were exposed to HyBlot CL Audoradiography Film (Denville, E3018) and developed using a Konica Minolta SRX-101A Medical Film Processor.

## RNAi

RNAi was performed using Lipofectamine RNAiMAX Transfection Reagent (Thermo Fisher Scientific, 13778150) with BLOCK-iT Fluorescent Oligo as a transfection control (Thermo Fisher Scientific, 13750062). *Plxn* knockdown was performed using Dharmacon ON-TARGET*plus* SMARTpool reagents with catalog numbers L-040789-01-0005, L-040790-01-0005, L-040791-01-0005, L-040980-00-0005, and L-056934-01-0005 for *Plxna1*, *Plxna2*, *Plxna3*, *Plxnb2*, and *Plxnd1*, respectively. *Nrp* oligos included *Nrp1*: GCACAAAUCUCUGAAACUA; and *Nrp2*: GACAAUGGCUGGACACCCA.

## RT-qPCR

NIH/3T3 cells were cultured as previously described and treated with low-serum media (0.5% bovine calf serum, 1% Penicillin–Streptomycin L-glutamine) containing SAG as indicated. RNA was isolated using the RNAqueous kit (Thermo Fisher Scientific, AM1912). cDNA was generated using 1 µg of template RNA (iScript RT Supermix, Bio-Rad, 1708841). cDNA was diluted 1:100, and qPCR was performed using SYBR green master mix (Thermo Fisher Scientific, AM9780) on an Applied BioSystems StepOnePlus Real-Time PCR System with the following primers: *Gli1* forward: GTGCACGTTTGA AGGCTGTC; *Gli1* reverse: GAGTGGGTCCGATTCTGGTG; *Ptch1* forward: GAAGCCACAGAAAACC CTGTC; *Ptch1* reverse: GCCGCAAGCCTTCTCTAGG; *Cyclophilin* forward: TCACAGAATTATTCCA GGATTCATG; and *Cyclophilin* reverse: TGCCGCCAGTGCCATT. *Cyclophilin* expression was used for normalization.

## Chicken in ovo neural tube electroporation

Electroporations were performed as previously described (*Tenzen et al., 2006*), using *Plxn*, *Smo^{M2}*, *Ptch^{ΔL2}*, and *Gli1* constructs cloned into the pCIG vector (*Megason and McMahon, 2002*). Briefly, DNA constructs (1.0 µg/µL total) were mixed with 50 ng/µL Fast green FCF dye (MilliporeSigma, F7252) and injected into the neural tube of Hamburger–Hamilton stage 11–13 chicken embryos (*Hamburger and Hamilton, 1951*). Embryos were dissected 48 hr post-injection and screened for GFP expression before being fixed in 4% PFA and prepared for immunofluorescent analyses. Embryos were embedded in Tissue-Tek OCT compound (Thermo Fisher Scientific, NC9806257), rapidly frozen over dry ice, and cryo-sectioned at a thickness of 12 microns using a Leica cryostat. Sections were affixed to glass slides and immunostained using the following antibodies: mouse IgG1 anti-PAX7 (1:20, Developmental Studies Hybridoma Bank [DSHB]), mouse IgG1 anti-NKX6.1 (1:20, DSHB), goat IgG anti-GFP (1:200, Abcam, ab6673), and rabbit IgG anti-MYC (1:100, Bethyl Laboratories, A190-205A). Slides were incubated with primary antibody overnight at 4°C followed by a 10 min DAPI stain (1:30,000 at room temperature, Invitrogen, D1306) and 1 hr incubation with secondary antibodies including Alexa Fluor 555 donkey anti-mouse IgG, Alexa Flour 488 donkey anti-goat IgG, and Alexa Flour 647 donkey anti-rabbit IgG (1:500, Invitrogen, A31570, A11055, A31573, respectively). Samples were visualized on a Leica Upright SP5X Light Laser Confocal Microscope, and figures were generated using Adobe Photoshop and Illustrator. The size of the NKX6.1 domain was measured using Adobe Illustrator in chicken neural tubes electroporated with pCIG (n = 6), *Gli1* (n = 4), and *Plxna1^{ΔECD}* (n = 17). These measurements were then normalized to the NKX6.1 domain size of the unelectroporated side of the neural tube.

## Mice

*Plxna1* (*Yoshida et al., 2006*) and *Plxna2* (*Suto et al., 2007*; *Duan et al., 2014*) mice, both on mixed genetic backgrounds, were generously provided by Dr. Alex Kolodkin. *Gli1^{lacZ}* animals were maintained on a mixed CD1 and C57BL/6J background (*Bai et al., 2002*). All mice were housed and cared for according to NIH guidelines, and all animal research was approved by the University of Michigan Medical School Institutional Animal Care and Use Committee. *Plxn* genotyping was performed using the following primers:

*Plxna1* WT_F: CCTGCAGATTGATGACGACTTCTG;
*Plxna1* WT_R: TCATGAGACCCAGTCTCCCTGTC;
*Plxna1* MT_F: GCATGCCTGTGACACTTGGCTCACT;
*Plxna1* MT_R: CCATTGCTCAGCGGTGCTGTCCATC;
*Plxna2* WT_F: GCTGGAACCATGTGAGAGCTGATC;
*Plxna2* WT_R: GGTCATCTAGTCGCAGGAGCTTGC;
*Plxna2* MT_F: GGTCATCTAGTCGCAGGAGCTTGC;
*Plxna2* MT_R: TACCCGTGATATTGCTGAAGAGCTTGG.

Ciliation frequency in *Plxna2^{+/-}* and *Plxna2^{-/-}* littermates was statistically assessed using a chi-squared analysis. Tissue preparation and X-gal, BrdU, and TUNEL staining were performed as previously described (*Duan et al., 2014*; *Holtz et al., 2015*; *Zhao et al., 2018*). Briefly, serial sagittal sections (16 µm) were collected from P7 brains and mounted onto six slides. One slide from each animal was used. BrdU labeling was carried out by injecting BrdU (20 mM, 50 mg/kg. Sigma B9285) intraperitoneally 1 hr prior to sacrifice (rat anti-BrdU, 1:500, Abcam ab6326). TUNEL staining was done following the manufacturer's protocol (Roche, REF-12156792910). Sections were imaged with a Zeiss Axio Observer Z1 equipped with a Zeiss Axiocam 503 mono camera and Zen software. Tiling and stitching were used to generate the BrdU and TUNEL images.

The total number of positive cells was quantified from four serial sections per slide to yield the average number of positive cells per animal; each data point represents a single animal.

## Acknowledgements

We are grateful to Dr AL Kolodkin (Johns Hopkins University, MD, USA) for providing *Plxn* constructs. Members of the Allen and Giger labs contributed technical assistance, insightful comments, and helpful suggestions. We are also thankful to Drs KS O'Shea, KJ Verhey, and JD Engel for sharing equipment and reagents. Confocal imaging was performed in the Microscopy Core at the University of

Michigan. We acknowledge the ENCODE consortium, and particularly the lab of Dr John Stamatoyannopoulous at the University of Washington for sharing their RNA-seq dataset on NIH/3T3 cells (GEO: GSM970853). JMP was supported by a Rackham Merit Fellowship, Benard Maas Fellowship, Bradley Merrill Patten Fellowship, Organogenesis Training Grant (T32 HD007505), and Ruth L Kirschstein National Research Service Award (F31 NS096734). RJG was supported by the Adelson Medical Foundation, Craig H Neilsen Foundation, and funding from the National Institutes of Health (R01 MH119346). BLA was supported by funding from the National Institutes of Health (R01 DC014428, R01 CA198074, and R01 GM118751). BLA and RJG were supported by an MCubed Research Grant from The University of Michigan.

## Additional information

### Funding

| Funder | Grant reference number | Author |
|---|---|---|
| National Institutes of Health | R01DC014428 | Benjamin L Allen |
| National Institutes of Health | R01CA198074 | Benjamin L Allen |
| National Institutes of Health | R01GM118751 | Benjamin L Allen |
| National Institutes of Health | R01MH119346 | Roman J Giger |
| National Institutes of Health | F31NS096734 | Justine M Pinskey |
| National Institutes of Health | T32HD007505 | Justine M Pinskey |

The funders had no role in study design, data collection and interpretation, or the decision to submit the work for publication.

### Author contributions

Justine M Pinskey, Tyler M Hoard, Conceptualization, Formal analysis, Validation, Investigation, Writing – original draft, Writing – review and editing; Xiao-Feng Zhao, Nicole E Franks, Formal analysis, Investigation; Zoë C Frank, Alexandra N McMellen, Investigation; Roman J Giger, Conceptualization, Resources, Formal analysis, Investigation, Writing – original draft, Writing – review and editing; Benjamin L Allen, Conceptualization, Resources, Formal analysis, Supervision, Funding acquisition, Investigation, Methodology, Writing – original draft, Project administration, Writing – review and editing

### Author ORCIDs

Justine M Pinskey ![ORCID] http://orcid.org/0000-0001-5656-5519
Tyler M Hoard ![ORCID] http://orcid.org/0000-0002-1193-0188
Xiao-Feng Zhao ![ORCID] http://orcid.org/0000-0002-7574-7163
Roman J Giger ![ORCID] http://orcid.org/0000-0002-2926-3336
Benjamin L Allen ![ORCID] http://orcid.org/0000-0003-2323-8313

### Ethics

All mice were housed in specific pathogen-free facilities at the University of Michigan. This study was approved by the University of Michigan Institutional Animal Care and Use Committee (IACUC; Protocol Number: PRO00010440).

### Decision letter and Author response

Decision letter https://doi.org/10.7554/eLife.74750.sa1
Author response https://doi.org/10.7554/eLife.74750.sa2

## Additional files

### Supplementary files
• Transparent reporting form

### Data availability
All data generated or analyzed during this study are included in the manuscript and supporting files.

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
