## [Editor Report]

This work demonstrates that Plexins, like their neuropilin-binding partners, promote HH signaling. The authors use both in vitro signaling assays, knockdown in chick neural tube patterning assays and PlexinA1 and A2 mutant mice to demonstrate that several Plexins enhance HH signaling in a way that depends on the Plexin GAP domain.

---

## [Decision Letter]

**Decision letter after peer review:**

Thank you for submitting your article "Plexins Promote Hedgehog Signaling Through Their Cytoplasmic GAP Activity" for consideration by *eLife*. Your article has been reviewed by 3 peer reviewers, one of whom is a member of our Board of Reviewing Editors, and the evaluation has been overseen by Didier Stainier as the Senior Editor. The following individual involved in review of your submission has agreed to reveal their identity: Phil Ingham (Reviewer #3).

Essential revisions:

It is curious that Plexins do not localize to cilia but require cilia for their activity in HH signaling. Do they promote ciliary localization of SMO, a key regulatory step in ciliary activation of GLI? Are primary cilia formed normally and present at normal frequency in cells with loss or over-expression of Plexins? This could help understand better how Plexins act to modulate the Hh pathway.

As the authors claim that the Plexina1r1ΔECD retains HH pathway augmenting capacity, they should use a statistical test to compare the activation to the pCIG control activation in Figure 3D, O and P. Does the ability of Plexina1r1ΔECD to augment HH pathway signaling, does PlexinA1 act partially through a GAP-independent activity or do the mutations not abrogate GAP activity?

In the chick neural tube experiments, how can the authors conclude that Plexin promotes Gli-dependent cellular responses since their data show that Plexin is not significantly affecting the fate (NKX6.1 and PAX7) of the cells? Quantitation of the NKX6.1 domain size revealed no significant differences between pCIG- and Plxna1ΔECD electroporated embryos (Figure S3J-M). Therefore, it is not appropriate to state that the authors "observed a minor shift in the NKX6.1 domain." Also, it would be relevant to know if ectopic cell migration can be caused by levels of Gli activity lower than those sufficient to induce Nkx6.1 expression.

The authors take the evidence that there are fewer Gli1-lacZ+ in the dentate gyruses of Plexin mutants to mean that "PLXNs can regulate HH pathway activation in vivo." However, decreased HH pathway activation is only one thing that could lead to decreased Gli1-lacZ+ cells in the dentate gyrus. In order to provide a convincing case for the role that Plexins play in Hh signaling in vivo, the in vivo Plexin loss of function experiments should be assessed in additional ways to Gli1-lacZ (Figure 6). Is there decreased proliferation of progenitors in the absence of Plexins? Is the dentate gyrus itself smaller? Is there increased death in the absence of Plexins? Excluding these other competing hypotheses will support the conclusion. Also, Is the hippocampal phenotype is enhanced in a Plxna1; Pxna2 double mutant?

The authors show that the effect of SmoM2 or Gli1 overexpression on Hh pathway activity can be potentiated by Plexins. They then conclude that "These data suggest that PLXNs function downstream of HH ligand at the level of GLI regulation…". It is unclear how this experiment allows them to conclude this, as the effect of Plexins could be downstream of Gli1, through the regulation of the transcription machinery, for example.

*Reviewer #1 (Recommendations for the authors):*

It is curious that Plexins do not localize to cilia but require cilia for their activity in HH signaling. Do they promote ciliary localization of SMO, a key regulatory step in ciliary activation of GLI?

As the authors claim that the Plexina1r1ΔECD retains HH pathway augmenting capacity, they should use a statistical test to compare the activation to the pCIG control activation in Figure 3D, O and P. Does the ability of Plexina1r1ΔECD to augment HH pathway signaling, does PlexinA1 act partially through a GAP-independent activity or do the mutations not abrogate GAP activity?

Quantitation of the NKX6.1 domain size revealed no significant differences between pCIG- and Plxna1ΔECD electroporated embryos (Figure S3J-M). Therefore, it is not appropriate to state that the authors "observed a minor shift in the NKX6.1 domain."

The authors take the evidence that there are fewer Gli1-lacZ+ in the dentate gyruses of Plexin mutants to mean that "PLXNs can regulate HH pathway activation in vivo." However, decreased HH pathway activation is only one thing that could lead to decreased Gli1-lacZ+ cells in the dentate gyrus. Is there decreased proliferation of progenitors in the absence of Plexins? Is the dentate gyrus itself smaller? Is there increased death in the absence of Plexins? Excluding these other competing hypotheses will support the conclusion.

*Reviewer #2 (Recommendations for the authors):*

This is interesting work that expands our knowledge of Hedgehog signaling. The work is well-done, well-written, and the figures are clear. I have comments that would help strengthen some of the experiments and improve the manuscript. In particular, the in vivo loss of function experiments could be measured in additional ways (using additional endpoints) to provide a convincing case of the role that Plexins play in Hh signaling in vivo.

Specific comments, in no particular order:

1. The authors show that the effect of SmoM2 or Gli1 overexpression on Hh pathway activity can be potentiated by Plexins. They then conclude that "These data suggest that PLXNs function downstream of HH ligand at the level of GLI regulation...". It is unclear to me how this experiment allows them to conclude this, as the effect of Plexins could be downstream of Gli1, through the regulation of the transcription machinery, for example.

2. Are primary cilia formed normally and present at normal frequency in cells with loss or over-expression of Plexins? This could help understand better how Plexins act to modulate the Hh pathway.

3. Are Gli1 protein levels affected by Plexins?

4. In order to provide a convincing case for the role that Plexins play in Hh signaling in vivo, the in vivo Plexin loss of function experiments should be assessed in additional ways to Gli1-lacZ (Figure 6). Also, proliferation should be measured (as previously shown to be Hh-dependent).

5. Data showing whether Plexins bind Shh (or not) should be presented.

6. The authors show that increased Plexin activity in chick neural tubes increases cell migration into the neural tube lumen. Is this effect of Plexins Gli-dependent?

7. In the chick neural tube experiments, how can the authors conclude that Plexin promotes Gli-dependent cellular responses since their data show that Plexin is not significantly affecting the fate (NKX6.1 and PAX7) of the cells? I was confused by this. The image shows a change, but the quantification does not.

8. Could loss of function experiments in chick neural tube using RNAi against multiple Plexins be performed? This would provide a very convincing case of the requirement of Plexins for Shh signaling.

*Reviewer #3 (Recommendations for the authors):*

1. In the introduction, the authors propose that functional redundancy can explain the failure to identify all modulators of Hh signalling activity in genetic screens. However, of the 10 examples that they mention, mutations in at least 5 have been isolated in mouse and or zebrafish on the basis of their Hh-related phenotypes (dispatched, hhip, scube2, ptch1/ptch2 and boc). It is also not clear that disp functions in a tissue or stage specific manner. I recommend that this section be revised.

2. The authors suggest that Plexins may modulate Hh signalling in a manner independent of NRP activity. This could be tested using the 3T3 system. It would also be interesting to assay the effects of NRP and SEMA in the absence of plexin function to address the questions raised in the Discussion.

3. For the in vivo analysis, it might have been more relevant to have analysed the Plxna3 mutant mouse. I would also like to know whether the hippocampal phenotype is enhanced in a Plxna1; Pxna2 double mutant.

4. Other points:

line 69: the abbreviation NRP is used for the first time without definition.

Figure 1B; the empty vector control, pCIG, is presented without explanation in the legend – in fact, it is only first mentioned in the text in line 267, towards the end of the Results section.

---

## [Author Response]

Essential revisions:It is curious that Plexins do not localize to cilia but require cilia for their activity in HH signaling. Do they promote ciliary localization of SMO, a key regulatory step in ciliary activation of GLI?

We thank the reviewers for this important suggestion. To address this point, we examined ciliary SMO localization in NIH/3T3 cells transfected either with a control vector (*pCIG*) or with *Plxna1* in the absence or presence of HH stimulation. We find that PLXNA1 expression does not alter baseline SMO cilia localization or SMO cilia localization in response to HH stimulation. These data are now included in a new supplemental figure (Figure 4—figure supplement 1A-D, F). See lines 274-276 of the revised manuscript for additional information.

Are primary cilia formed normally and present at normal frequency in cells with loss or over-expression of Plexins? This could help understand better how Plexins act to modulate the Hh pathway.

We agree with the reviewers that this is a key point to address. As part of the above-described experiment in NIH/3T3 cells, we assessed cilia number and cilia length in the presence or absence of HH stimulation and in the context of PLXNA1 expression. Notably, no changes in primary cilia frequency or length are detected in these cells (see Figure 4—figure supplement 1E, G). See lines 274-279 of the revised manuscript for additional information.

We also assessed cilia number and cilia length in P7 *Plxna2* mutant animals. Despite significant reductions in *Gli1* expression in the hippocampi of these animals, we do not detect any changes in ciliation frequency or cilia length (see Figure 4—figure supplement 1H, I). See lines 276-279 of the revised manuscript for additional information.

As the authors claim that the Plexina1r1ΔECD retains HH pathway augmenting capacity, they should use a statistical test to compare the activation to the pCIG control activation in Figure 3D, O and P.

We agree with the reviewers and have added the requested p-values to Figure 3D, O, P and Figure 3—figure supplement 2B.

Does the ability of Plexina1r1ΔECD to augment HH pathway signaling, does PlexinA1 act partially through a GAP-independent activity or do the mutations not abrogate GAP activity?

We thank the reviewers for this excellent question. The *Plxna1r1* mutation affects only two amino acids, leaving the rest of the extensive PLXN cytoplasmic domain unaltered. It is therefore possible that PLXNs could mediate HH signaling partially through a GAP-independent mechanism, particularly given the vast network of PLXN cytoplasmic binding partners (Hota and Buck 2012). As the reviewers suggest, it is also possible that the *Plxna1r1* mutation does not fully abrogate GAP activity. To address these possibilities, we have made several additional mutations in *Plxna1* and tested their effects on HH promotion (see Figure 3—figure supplement 2). First, we mutagenized a second conserved arginine residue (R2) also demonstrated to be essential for GAP-dependent PLXN activation during Semaphorin signaling (Rohm et al. 2000). Mutating both the R1 and R2 sites to alanine should abrogate any residual PLXN GAP activity. Notably, PLXN proteins with both mutations still partially promote HH pathway activity (Figure 3—figure supplement 2B). These data suggest that other cytoplasmic determinants contribute to PLXN-mediated HH promotion. Previous work identified two FYN kinase phosphorylation sites as key mediators of PLXN function (St Clair et al. 2018). We find that mutation of one of these tyrosine residues (but not the other) can significantly abrogate PLXN-mediated HH pathway activation. Notably, mutation of both the GAP domain and the FYN kinase phosphorylation site rendered PLXN inert in the context of HH signal transduction (Figure 3—figure supplement 2C). See lines 193-208 of the revised manuscript for additional information.

In the chick neural tube experiments, how can the authors conclude that Plexin promotes Gli-dependent cellular responses since their data show that Plexin is not significantly affecting the fate (NKX6.1 and PAX7) of the cells?

The reviewers raise an important point. In our initial manuscript, we observed that constitutive SMO activity promotes ectopic NKX6.1 expression and loss of PAX7. However, constitutive GLI activator expression results in an additional phenotype– ectopic cell migration in the dorsal neural tube. Thus, HH pathway activation can result in both ectopic cell fate specification and ectopic cell migration. Notably, *Plxna1*^D*ECD*^ expression mimics the migratory phenotype, but not the cell specification phenotype. A key question is whether the PLXN-mediated migratory phenotype is HH-dependent. To address this, we co-electroporated neural tubes with *Plxna1*^D*ECD*^ and *Ptch1^∆L2^*, a constitutively active form of *Ptch1* that is insensitive to HH ligands. Strikingly, we find that *Ptch1^∆L2^* abrogates the PLXN-mediated migratory phenotype, suggesting that increased PLXN-mediated migration in the neural tube is HH-dependent. We now include these data as a new Figure 6 and have revised the manuscript accordingly (see lines 309-318).

Quantitation of the NKX6.1 domain size revealed no significant differences between pCIG- and Plxna1ΔECD electroporated embryos (Figure S3J-M). Therefore, it is not appropriate to state that the authors "observed a minor shift in the NKX6.1 domain."

We apologize for the confusion, and have revised the results (lines 296-302) to clarify our findings.

Also, it would be relevant to know if ectopic cell migration can be caused by levels of Gli activity lower than those sufficient to induce Nkx6.1 expression.

As noted above, constitutive SMO activation and constitutive GLI activator both induce ectopic cell fate specification. However, only GLI activator induces ectopic cell migration (see Figure 5). These data suggest that either a higher level of GLI activity is necessary to induce migration (compared to cell fate specification), or that the cell fate specification and cell migration responses represent distinct outcomes due to which GLI transcription factors are engaged (i.e., primarily *GLI2* in embryos with constitutive SMO activation versus GLI1 in embryos with GLI1 electroporation). While we cannot distinguish between these possibilities (or others), the demonstration that *Ptch1^∆L2^* can block PLXN-mediated migration (see Figure 6) suggests that regardless of mechanism, this is a HH-dependent outcome. To fully explore the nature of GLI transcription factor activity in response to PLXN expression would represent an entirely new avenue of research.

The authors take the evidence that there are fewer Gli1-lacZ+ in the dentate gyruses of Plexin mutants to mean that "PLXNs can regulate HH pathway activation in vivo." However, decreased HH pathway activation is only one thing that could lead to decreased Gli1-lacZ+ cells in the dentate gyrus. In order to provide a convincing case for the role that Plexins play in Hh signaling in vivo, the in vivo Plexin loss of function experiments should be assessed in additional ways to Gli1-lacZ (Figure 6). Is there decreased proliferation of progenitors in the absence of Plexins? Is the dentate gyrus itself smaller? Is there increased death in the absence of Plexins? Excluding these other competing hypotheses will support the conclusion.

The reviewers raise several important points. We have experimentally addressed this by analyzing BrdU incorporation and TUNEL staining in both *Plxna1* and *Plxna2* mutant animals. Notably, we find no significant differences in either BrdU incorporation (as a measure of cell proliferation) or TUNEL staining (as a measure of cell death) in either mutant line (see Figures S7 and S8). These data suggest that the reduction in Gli1+ cells is not due to secondary effects from reduced proliferation or increased apoptosis, supporting our conclusion that PLXNs do indeed regulate HH pathway activation in vivo. See lines 335-339 of the revised manuscript for additional information.

Also, Is the hippocampal phenotype is enhanced in a Plxna1; Pxna2 double mutant?

While the reviewers raise an interesting question, unfortunately *Plxna1;Plxna2* double mutant animals display embryonic lethality. Thus, we are not able to answer the reviewers’ question regarding enhanced hippocampal phenotypes. We have added this information to the text (see lines 339-341).

The authors show that the effect of SmoM2 or Gli1 overexpression on Hh pathway activity can be potentiated by Plexins. They then conclude that "These data suggest that PLXNs function downstream of HH ligand at the level of GLI regulation…". It is unclear how this experiment allows them to conclude this, as the effect of Plexins could be downstream of Gli1, through the regulation of the transcription machinery, for example.

The reviewers are correct that PLXNs could function downstream of GLI1, acting to generally enhance transcription. To directly test this, we again employed luciferase assays to measure potential PLXN-mediated transcriptional effects on a separate pathway; specifically, we used a reporter construct containing multiple TCF/LEF binding sites (TOP-FLASH) to measure Wnt pathway activity (Molenaar et al. 1996). Surprisingly, we find that PLXNA1 does not promote Wnt pathway activation. Instead, PLXNA1, and to a greater degree PLXNA1^DECD^, inhibits Wnt pathway activity, with PLXNA1^DECD^ reducing Wnt pathway activity to baseline levels (see Figure 3—figure supplement 3). These data suggest that PLXN does not act to generally promote transcription, and instead has opposing consequences on HH and Wnt transcriptional readouts. See lines 215-225 of the revised manuscript for additional information.

Reviewer #2 (Recommendations for the authors):This is interesting work that expands our knowledge of Hedgehog signaling. The work is well-done, well-written, and the figures are clear. I have comments that would help strengthen some of the experiments and improve the manuscript. In particular, the in vivo loss of function experiments could be measured in additional ways (using additional endpoints) to provide a convincing case of the role that Plexins play in Hh signaling in vivo.Specific comments, in no particular order:1. The authors show that the effect of SmoM2 or Gli1 overexpression on Hh pathway activity can be potentiated by Plexins. They then conclude that "These data suggest that PLXNs function downstream of HH ligand at the level of GLI regulation...". It is unclear to me how this experiment allows them to conclude this, as the effect of Plexins could be downstream of Gli1, through the regulation of the transcription machinery, for example.

We thank the reviewer for their favorable assessment and appreciate their recommendations to add additional in vivo loss of function experiments, which are addressed in the response to Essential revisions above.

2. Are primary cilia formed normally and present at normal frequency in cells with loss or over-expression of Plexins? This could help understand better how Plexins act to modulate the Hh pathway.

See response to Essential revisions above.

3. Are Gli1 protein levels affected by Plexins?

See response to Essential revisions above and Figure 4- figure supplement 1H.

4. In order to provide a convincing case for the role that Plexins play in Hh signaling in vivo, the in vivo Plexin loss of function experiments should be assessed in additional ways to Gli1-lacZ (Figure 6). Also, proliferation should be measured (as previously shown to be Hh-dependent).

We have not directly examined GLI1 protein levels. Future studies will investigate the consequence of PLXNs on levels, processing and localization of all GLI proteins based on the findings from this study.

5. Data showing whether Plexins bind Shh (or not) should be presented.

See response to Essential revisions above, Figure 7—figure supplement 1 and Figure 7—figure supplement 1.

6. The authors show that increased Plexin activity in chick neural tubes increases cell migration into the neural tube lumen. Is this effect of Plexins Gli-dependent?

The reviewer raises an interesting point. However, the data with the *Plxna1^∆ECD^* construct, which lacks the entire extracellular domain suggests that PLXN binding to SHH is not required for HH pathway promotion (see Figure 3). Instead, our experiments suggest that PLXN functions downstream of HH ligand (see Figure 3).

7. In the chick neural tube experiments, how can the authors conclude that Plexin promotes Gli-dependent cellular responses since their data show that Plexin is not significantly affecting the fate (NKX6.1 and PAX7) of the cells? I was confused by this. The image shows a change, but the quantification does not.

See response to Essential revisions above and Figure 3-figure supplement 3.

8. Could loss of function experiments in chick neural tube using RNAi against multiple Plexins be performed? This would provide a very convincing case of the requirement of Plexins for Shh signaling.

While we appreciate the reviewer’s suggestion, this experiment would be technically very challenging, given that several PLXNs are expressed in the chicken neural tube (Mauti et al. 2006), and we would likely need to achieve robust knockdown of multiple *Plxns* to reveal a phenotype. Instead, we have relied on knockdown approaches in cell culture and genetic deletion in mice to assess the consequences of PLXN loss-of-function on HH signaling.

Reviewer #3 (Recommendations for the authors):1. In the introduction, the authors propose that functional redundancy can explain the failure to identify all modulators of Hh signalling activity in genetic screens. However, of the 10 examples that they mention, mutations in at least 5 have been isolated in mouse and or zebrafish on the basis of their Hh-related phenotypes (dispatched, hhip, scube2, ptch1/ptch2 and boc). It is also not clear that disp functions in a tissue or stage specific manner. I recommend that this section be revised.

We have revised this section according to the reviewer’s recommendations (see lines 47-51).

2. The authors suggest that Plexins may modulate Hh signalling in a manner independent of NRP activity. This could be tested using the 3T3 system. It would also be interesting to assay the effects of NRP and SEMA in the absence of plexin function to address the questions raised in the Discussion.

We thank the reviewer for this suggestion and agree that it would be interesting to assess HH signaling via Plexins in the absence of Neuropilins and vice versa. We have added a line in the discussion to include these experiments as suggestions for future studies (see lines 405-406).

3. For the in vivo analysis, it might have been more relevant to have analysed the Plxna3 mutant mouse. I would also like to know whether the hippocampal phenotype is enhanced in a Plxna1; Pxna2 double mutant.

We agree with the reviewer that analyzing Plxna3 mutant animals would be valuable as would analyzing *Plxna1;Plxna2* double mutant animals. Unfortunately, we did not have access to *Plxna3* mutant mice at the time of our experiments, and *Plxna1;Plxna2* double mutant animals are embryonic lethal.

References

Hota PK, Buck M. 2012. Plexin structures are coming: opportunities for multilevel investigations of semaphorin guidance receptors, their cell signaling mechanisms, and functions. *Cell Mol Life Sci* 69: 3765-3805.

Mauti O, Sadhu R, Gemayel J, Gesemann M, Stoeckli ET. 2006. Expression patterns of plexins and neuropilins are consistent with cooperative and separate functions during neural development. *BMC Dev Biol* 6: 32.

Molenaar M, van de Wetering M, Oosterwegel M, Peterson-Maduro J, Godsave S, Korinek V, Roose J, Destree O, Clevers H. 1996. XTcf-3 transcription factor mediates β-catenin-induced axis formation in *Xenopus* embryos. *Cell* 86: 391-399.

Rohm B, Ottemeyer A, Lohrum M, Puschel AW. 2000. Plexin/neuropilin complexes mediate repulsion by the axonal guidance signal semaphorin 3A. *Mech Dev* 93: 95-104.

St Clair RM, Emerson SE, D'Elia KP, Weir ME, Schmoker AM, Ebert AM, Ballif BA. 2018. Fyn-dependent phosphorylation of PlexinA1 and PlexinA2 at conserved tyrosines is essential for zebrafish eye development. *FEBS J* 285: 72-86.